# TrkA-mediated endocytosis of p75-CTF prevents cholinergic neuron death upon γ-secretase inhibition

María Luisa Franco[1],*, Irmina García-Carpio[1],*, Raquel Comaposada-Baró[1] , Juan J Escribano-Saiz[1], Lucía Chávez-Gutiérrez[2], Marçal Vilar[1]

γ-secretase inhibitors (GSI) were developed to reduce the generation of Aβ peptide to find new Alzheimer's disease treatments. Clinical trials on Alzheimer's disease patients, however, showed several side effects that worsened the cognitive symptoms of the treated patients. The observed side effects were partially attributed to Notch signaling. However, the effect on other γ-secretase substrates, such as the p75 neurotrophin receptor (p75NTR) has not been studied in detail. p75NTR is highly expressed in the basal forebrain cholinergic neurons (BFCNs) during all life. Here, we show that GSI treatment induces the oligomerization of p75CTF leading to the cell death of BFCNs, and that this event is dependent on TrkA activity. The oligomerization of p75CTF requires an intact cholesterol recognition sequence (CRAC) and the constitutive binding of TRAF6, which activates the JNK and p38 pathways. Remarkably, TrkA rescues from cell death by a mechanism involving the endocytosis of p75CTF. These results suggest that the inhibition of γ-secretase activity in aged patients, where the expression of TrkA in the BFCNs is already reduced, could accelerate cholinergic dysfunction and promote neurodegeneration.

## Introduction

Alzheimer's disease (AD) is characterized by cognitive deficits and is one of the most commonly diagnosed types of dementia. Amyloid plaques are one of the neuropathological hallmarks of AD and are comprised of misfolded Aβ peptides. Aβ peptides are generated by sequential cleavage of the amyloid precursor protein (APP) by the β- and the γ-secretases. Mutations in the γ-secretase and APP cause autosomal dominant, early onset AD (De Strooper & Chávez Gutiérrez, 2015). Owing to its involvement in the production of Aβ production and close link to AD pathogenesis, γ-secretases have been considered to be one of the most promising targets as AD therapeutics. The development of γ-secretase inhibitors (GSIs) was in fact an area holding great expectations. GSIs were used in clinical trials to reduce the production of Aβ in AD patients. The GSI semagacestat (LY450139) Phase 3 clinical trial (Hopkins, 2010) was stopped because of adverse effects (such as increased risk of skin cancer) and a worsening of memory in the GSI treated group (Doody et al, 2013). The main reason of such failure likely relies on the fact that γ-secretases do not only process APP but also cleave many other type 1 transmembrane proteins (De Strooper & Chávez Gutiérrez, 2015), and thus, the concomitant GSI-mediated inhibition of the cleavage of other substrates of γ-secretase likely caused the observed undesirable consequences. The inhibition of the cleavage of Notch received great attention (Olsauskas-Kuprys et al, 2013; De Strooper, 2014); however, the impact that semagacestat could have had on other γ-secretase substrates is unclear. Although essential during development, Notch function in the adult central nervous system (CNS) is highly restricted to the population of neural stem cells and probably other substrates could better explain the worsening of the cognitive function seen in the clinical trial. One of the physiologically relevant substrates of γ-secretase in the brain is the p75 neurotrophin receptor. The p75 neurotrophin receptor (p75NTR) is a member of the TNF receptor superfamily (Ibáñez & Simi, 2012; Bothwell, 2014), and it is best known by its role in programmed neuronal death during development or in response to injury in the adult brain (Ibáñez & Simi, 2012). It also regulates axonal growth and synaptic plasticity, as well as cell proliferation, migration, and survival (Kraemer et al, 2014; Vilar, 2017). These functions can be elicited by the association of p75NTR with different ligands and co-receptors leading to the activation of various signaling pathways (Roux & Barker, 2002). Importantly, p75NTR is highly expressed in the basal forebrain cholinergic neurons (BFCNs) during all stages of their development, a neuronal population well known for their involvement of complex cognitive tasks via their innervation to the cortex and hippocampus.

p75NTR undergoes regulated intramembrane proteolysis (RIP) (Kanning et al, 2003; Jung et al, 2003), a two-step process that

[1]Molecular Basis of Neurodegeneration Unit, Institute of Biomedicine of València (IBV-CSIC), València, Spain [2]Vlaams Instituut voor Biotechnologie Katholieke Universiteit (VIB-KU) Leuven Center for Brain and Disease, Leuven, Belgium

Correspondence: mvilar@ibv.csic.es
Irmina García-Carpio's present address is Division of Developmental Immunology, Biocenter, Medical University of Innsbruck, Innsbruck, Austria
*María Luisa Franco and Irmina García-Carpio contributed equally to this work

involves the sequential cleavage of p75$^{NTR}$ by the $\alpha$- and $\gamma$-secretases (Fig 1A). The $\alpha$-secretase activity is mediated by TACE/ADAM-17, a member of the A Disintegrin And Metalloprotease (ADAM) family (Weskamp et al, 2004; Bronfman, 2007) and generates a C-terminal membrane–anchored fragment (p75-CTF). In vivo p75$^{NTR}$ shedding was described for the first time in Schwann cells after axotomy (DiStefano & Johnson, 1988). In vitro, p75$^{NTR}$ shedding is induced by protein kinase C activators, such as phorbol esters (Kanning et al, 2003), or by the activation of TrkA (Urra et al, 2007; Ceni et al, 2010). The p75-CTF is further processed by the $\gamma$-secretase that cleaves the transmembrane domain between Val264 and Val265 to release a soluble intracellular fragment (ICD) (Jung et al, 2003; Kanning et al, 2003). Moreover, overexpression of p75-CTF in a form that cannot be processed by $\gamma$-secretase has been proven to promote cell death in neurons, indicating that p75-CTF processing and clearance from the membrane relies on $\gamma$-secretase activity (Coulson et al, 2008). Of note, covalent p75NTR dimerization, through the evolutionary conserved transmembrane cysteine residue, present in its transmembrane domain (Vilar et al, 2009b; Nadezhdin et al, 2016), is required for the induction of cell death upon stimulation by pro-neurotrophins in vitro and in vivo (Vilar et al, 2009b; Tanaka et al, 2016).

Here, we found that $\gamma$-secretase activity processes p75-CTF only in monomeric status, suggesting that the dimerization/oligomerization of p75-CTF represents a mechanism that regulates its clearance from the membrane. Interestingly, we show that the inhibition of $\gamma$-secretase increases the levels of p75-CTF and promotes the formation of p75-CTF oligomers which in turn leads to exacerbated toxicity. Finally, we demonstrate that the activation of TrkA abolishes p75-CTF oligomerization and protects from cell death by promoting the endocytosis of p75CTF. In conclusion, our results reveal a novel mechanism underlying the RIP of p75, where the oligomerization of the receptor (substrate) and its subcellular location protects it from $\gamma$-secretase–mediated processing and exacerbates its deadly function.

## Results

### p75-CTF disulfide dimerization contributes to cell death

RIP of p75$^{NTR}$ is required for signaling. Endogenously, generation of CTF correlates with rapid p75$^{NTR}$-mediated apoptosis upon injury conditions (Coulson et al, 2000; Sotthibundhu et al, 2008). To determine the contribution of different protein degradation pathways to the turnover of p75$^{NTR}$ upon RIP, we took advantage of a truncated p75$^{NTR}$ construct that mimics the endogenously generated CTF (p75-CTF) and performed cycloheximide chase experiments in HeLa cells (Fig 1B). Consistently with previous reports (Kanning et al, 2003), proteasomal inhibition with epoxomicin caused an accumulation of p75-ICD, the later was abolished by treatment with GSIs compound E (CE, Fig 1B) or semagacestat, SG (Fig S1A). Interestingly, the treatment with autophagy (wortmannin, W) (Fig 1B) and lysosomal inhibitors (NH$_4$Cl) (Fig S1A) did not affect p75-CTF turnover or p75-ICD stability in HeLa cells (Fig 1C). Our results were recapitulated in the endogenous p75$^{NTR}$ from PC12 cells (Fig S1B) thus supporting the conclusion that p75-CTF processing and clearance from the membrane relies on $\gamma$-secretase activity.

Next, we turned our attention to the p75-CTF activity. The mechanism underlying p75-CTF toxicity and its role in cell death

induction is not fully understood. p75$^{NTR}$ dimerization through the Cys$^{257}$ has been described as an essential process for p75$^{NTR}$-mediated cell death in response to neural insults (Vilar et al, 2009b; Tanaka et al, 2016). Here, we assessed p75$^{NTR}$-CTF dimerization in conditions that elevate its steady-state levels. First, we evaluated the contribution of Cys$^{257}$ in the formation of these dimers and in p75$^{NTR}$ CTF processing by $\gamma$-secretase. HeLa cells were transfected to express the p75-CTF fragment or a mutated version unable to form disulfide covalent bonds (p75-CTF-C257A). Cultures were then incubated overnight in the presence of the GSI CE to induce CTF accumulation (lanes indicated as 16 h in Fig 1D and E). Given that substrate recognition and processing by $\gamma$-secretase takes place in the intramembranous space, we assessed $\gamma$-secretase function in total membrane fractions. This is an alternative and well-validated cell-free system for the study of $\gamma$-secretase activity (Chávez-Gutiérrez et al, 2008). Overnight inhibition of $\gamma$-secretase activity prevented CTF degradation and resulted in the accumulation of p75-CTF dimers of the wt fragment but not of the C257A mutant (lanes 3, 4 and 7, 8 in Fig 1D and E). This indicates that p75-CTF dimerizes through the transmembrane Cys$^{257}$ and furthermore, it does it in a concentration-dependent manner. Isolated membranes from untreated or inhibitor-treated cells were incubated at 37°C for 1 h in absence or presence of CE (indicated as −/+ in Fig 1D and E, respectively) and analyzed under nonreducing conditions by Western Blot. Incubation of non-GSI treated membrane fractions at 37°C showed that endogenous $\gamma$-secretase was able to process both wt and C257A p75-CTF substrates in similar extents. Remarkably, p75-CTF accumulation after 16 h of GSI/CE treatment promoted dimer formation, but the elevation in the $\gamma$-secretase substrate did not lead to an increase in the generation of p75-ICD, suggesting that the p75$^{NTR}$ CTF dimers are not contributing to the generation of p75-ICD (Fig 1D, lanes 1 and 3).

Next, we analyzed P2 mice isolated dorsal root ganglia (DRG) neurons that express endogenous levels of p75$^{NTR}$, treated for 24 h with the GSI CE. Nonreducing SDS–PAGE Western blotting showed that, upon $\gamma$-secretase inhibition, the endogenously produced p75-CTF accumulates as monomers and dimers (Fig 1F). This demonstrates that p75-CTF dimers are formed from endogenous levels in neurons undergoing $\gamma$-secretase inhibition.

We next transfected mouse cortical neurons, which do not express significant levels of p75$^{NTR}$ in cell culture, with different p75$^{NTR}$ constructs to analyze apoptosis induction by caspase-3 cleavage (Fig 1G and H). Overexpression of p75$^{NTR}$ constructs (p75$^{NTR}$ full length or CTF, Fig 1I) induced a significant increase in cell death in the presence of CE. Of note, this cell death correlates with CTF accumulation upon CE treatment, as it is shown by immunoprecipitation of the different p75$^{NTR}$ constructs transfected (Fig 1I). This supports the idea that the p75-CTF by itself is sufficient to mediate apoptosis. The significant increment in cell death observed in p75-CTF-wt overexpressing neurons relative to the mutant, suggests a partial, but significant, contribution of the cysteine residue to the p75-CTF–mediated toxicity.

### p75-CTF disulfide dimers decrease protein turnover

To further understand the role of the Cys$^{257}$ in p75-CTF–mediated cell death upon $\gamma$-secretase inhibition, we analyzed the stability of p75-CTF-C257A in a cycloheximide chase experiment (Fig 2A). Cycloheximide blocks protein synthesis in eukaryotic cells and

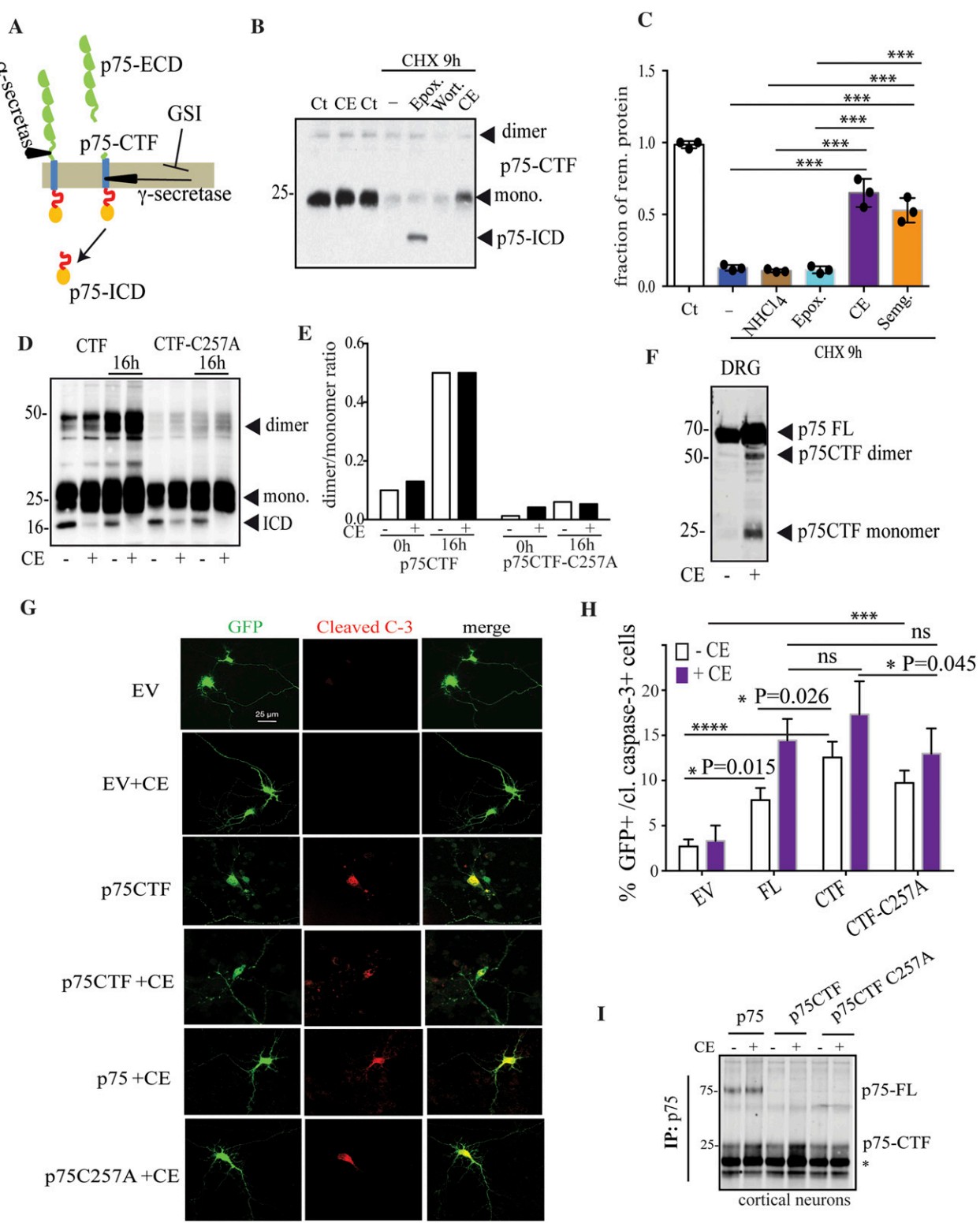

**Figure 1.  γ-secretase inhibition drives to p75CTF dimerization and cell death.**
**(A)** Schematic overview of p75 regulated intramembrane proteolysis by α- and γ-secretases. **(B, C)** Expression levels of p75-CTF in HeLa cells transiently transfected with HA-p75-CTF and incubated with cycloheximide (5 μg/ml) for 9 h in the absence or presence of inhibitors of different protein degradation pathways (see text). Western blotting and densitometric analysis of p75-CTF remaining protein indicate that its turnover is mostly mediated by γ-secretase activity and not by lysosomal or proteasomal degradation. To determine the specific activity for the different inhibitors, p75-CTF levels were normalized with respect to transfected untreated cells (Ct). **(D)** de novo p75-ICD generation from purified total membranes prepared from HeLa cells transfected with the indicated constructs and detected with p75^NTR antibody. Transfected

accordingly enables to estimate the protein half-life. Quantification of the turnover rates showed that wt p75-CTF presents a significant higher half-life than the mutant p75-CTF-C257A, and this difference increases in the presence of the GSI (two-way ANOVA analysis; time factor $F(3, 6) = 970.9$ $P < 0.0001$; mutant factor $F(3, 6) = 59.22$, $P < 0.0001$; both factors $F(9, 18) = 60.1$ $P < 0.0001$) (Fig 2B). Interestingly, the analysis of cycloheximide chase experiments under nonreducing conditions revealed that p75-CTF dimers were resistant to degradation over time and suggested that their formation decreases protein turnover (Fig 2C).

### TrkA promotes p75CTF internalization in a ligand-dependent manner

It has been shown that p75-CTF is internalized and cleaved by γ-secretase in endosomes (Urra et al, 2007). Therefore, we wondered whether the covalent dimer formation played a role in its internalization and turnover. To address this point, HEK293 cells were transfected with different HA-tagged p75-CTF constructs and incubated at 37°C for 0, 5, 15, 30, 60, and 120 min for different time points (Fig 2D). Analysis of p75-CTF location by immunofluorescence showed a significant faster constitutive internalization of p75-CTF-C257A proteins compared with p75-CTF (two-way ANOVA analysis; time factor $F(5, 10) = 335.3$ $P < 0.0001$; mutant factor $F(3, 6) = 46.91$, $P = 0.0001$; both factors $F(11, 74) = 15.30$ $P < 0.0001$) (Fig 2E).

In several neuronal cell types p75NTR is co-expressed with one member of the Trk family. TrkA is usually co-expressed with p75NTR in sympathetic and DRG neurons as well as in the PC12 cell line. To study the role of TrkA in p75CTF internalization, we transfected HEK293 with TrkA and p75-CTF or TrkA and p75-CTF-C257A and quantify the internalization of the CTF after NGF stimulation. Quantification of Fig 2D shows that in the presence of TrkA, the wt p75-CTF is more rapidly internalized upon stimulation with NGF, relative to the p75-CTF alone (no TrkA) or to TrkA but no NGF stimulation (Fig 2E). The mutant p75-CTF-C257A is internalized more slowly in the presence of TrkA than in its absence (Fig 2D and E). Inhibition of TrkA activity with the Trk-specific inhibitor K-252a or with amiloride, a specific inhibitor of macropinocytosis, inhibits p75CTF internalization (Fig 2F). To further prove the role of TrkA in p75CTF internalization, we co-transfected HA-p75-CTF and TrkA in 293T cells and measured the percentage of cells expressing surface p75CTF before and after incubation with NGF. Flow cytometry analysis of HA-positive cells, showed a decrease in cells presenting p75CTF at the surface upon co-expression with TrkA, in a ligand-dependent manner (Fig S2A and B). These results indicate that the internalization of p75CTF is promoted by NGF-mediated activation of TrkA, and not just by TrkA expression.

To further support that TrkA is able to mediate the internalization of p75CTF we transfected PC12 (expressing endogenous TrkA levels) and PC12nnr5 cells (that do not express TrkA) with the HA-tagged wt p75CTF construct. We then quantified wt p75CTF internalization upon NGF stimulation. As shown in Fig 2G and H, NGF triggered p75CTF internalized in the PC12 cells line after 60 min, but the treatment did not cause p75CTF internalization in the PC12nnr5 cell line in the same time frame.

Altogether, these results indicate that the NGF-mediated activation of TrkA promotes p75CTF internalization in a ligand-dependent manner.

### p75-CTF dimers are not cleaved by γ-secretase

Our initial findings suggested that endogenous γ-secretase was not processing naturally occurring covalent dimers of p75-CTF (Fig 1D). To further analyze these findings, we benefit from an in vitro approach, where the purified substrate and enzyme are used. Given that γ-secretase substrate recognition and cleavage takes place in the intra-membranous space, we generated different constructs containing the p75NTR transmembrane domain and juxtamembrane region, fused to a C-terminus triple FLAG tag sequence that facilitate its purification and detection, C101-p75-wt-3xFlag (Fig 3A). These constructs are reminiscent to the C99 construct generated by the β-secretase activity from APP, C99-APP-3xFlag (Chávez-Gutiérrez et al, 2008). Western blot analyses of the purified wt and C257A mutant peptides showed that C101-p75-wt, but not C101-p75-C257A, form DTT-sensitive disulfide dimers (Fig 3B).

We then analyzed the γ-secretase–mediated processing of the wild-type and mutant C101 fragments by co-incubating them with purified γ-secretase complex (Chávez-Gutiérrez et al, 2012). Monitoring of the γ-secretase cleavage was followed by quantification of the c-terminal fragment product (ΔICD-3xFLAG) levels. As γ-secretase activity positive control, we followed the endoproteolytic cleavage of C99-APP-x3FLAG (Fig 3C). The presence of the GSI X (a transition state-analogue) inhibited the generation of the ΔICD-3xFLAG, demonstrating the specificity of the proteolytic reactions. Remarkably, the dimeric C101-p75-wt substrate, but not the monomeric C101-p75 and C101-p75-C257A, was resistant to γ-secretase cleavage. Furthermore, whereas DTT treatment of the mutant C257A C101-p75 substrate did not affect γ-secretase activity, the

---

cells were first incubated overnight (16 h) in the presence or absence of CE (10 μM) to induce CTF accumulation. Overnight γ-secretase inhibition unequivocally drives to p75-CTF dimerization in a concentration-dependent manner. Total membranes from these cells were purified and incubated for 1 h at 37°C in the presence or absence of CE (indicated as +/− in the figure). Membrane lysates analyzed under nonreducing conditions show specific p75-ICD generation by endogenous γ-secretase in both wt and mutant C257A p75-CTF. **(E)** Quantification of the Western blot showing the ratio dimer/monomer of p75CTF in the different conditions (n = 1). **(F)** Dorsal root ganglia from P2 mice were isolated and incubated for 24 h at 37°C in the presence or absence of CE (10 μM). Lysate analysis under no reducing conditions demonstrates p75CTF endogenous dimerization. **(G)** Representative microscopy images show caspase-3 cleavage immunostaining of isolated mice cortical neurons transiently overexpressing control GFP or p75-CTF and GFP after 48-h incubation at 37°C with CE (10 μM). **(H)** Apoptotic cell death was determined for the expression of the different p75NTR constructs as the percentage of GFP caspase-3–cleaved positive cells in the presence or absence of CE. Cell death in transfected cortical neurons shows a small but significant reduction in the p75-CTF-C27A mutant with respect to the wt. **(I)** Western blot of the p75-specific antibody immunoprecipitated lysates from transfected cortical neurons with the indicated constructs. * Immunoglobulin light chain. The data are shown as mean ± SD, N = 3 independent experiments. Per experiment, all GFP+ neurons per well were counted. In total, more than 50 transfected neurons were quantified per condition. **(C, G)** t test (C) and two-way ANOVA followed by Tukey's post-test (G) were used to determine the statistical significance, *$P < 0.05$, **$P < 0.01$, ***$P < 0.001$, ****$P < 0.0001$.
Source data are available for this figure.

**Figure 2. Cysteine-257 increases p75CTF internalization and turnover.**

**(A)** Representative SDS–PAGE/Western blot from cycloheximide chase analysis of HeLa cells transiently expressing wt or mutant p75-CTF, in the absence or presence of CE (10 μM). **(B)** Quantification of the effect of p75 Cys257 substitution in γ-secretase processivity of CTF. p75-CTF remaining protein was calculated with respect to untreated cells (0 h) to determine its processing over time. **(C)** Mutation of Cys at position 257 decreases p75-CTF stability. Nonreducing SDS and Western blotting analysis of cycloheximide chase experiment in transfected HeLa cells reveal that p75-CTF dimers are resistant to degradation over time. Arrowheads indicate p75-CTF monomers' and dimers' position. **(D)** Representative confocal images of kinetic internalization experiments in HEK293 cells expressing wt or mutant HA-p75-CTF constructs in the

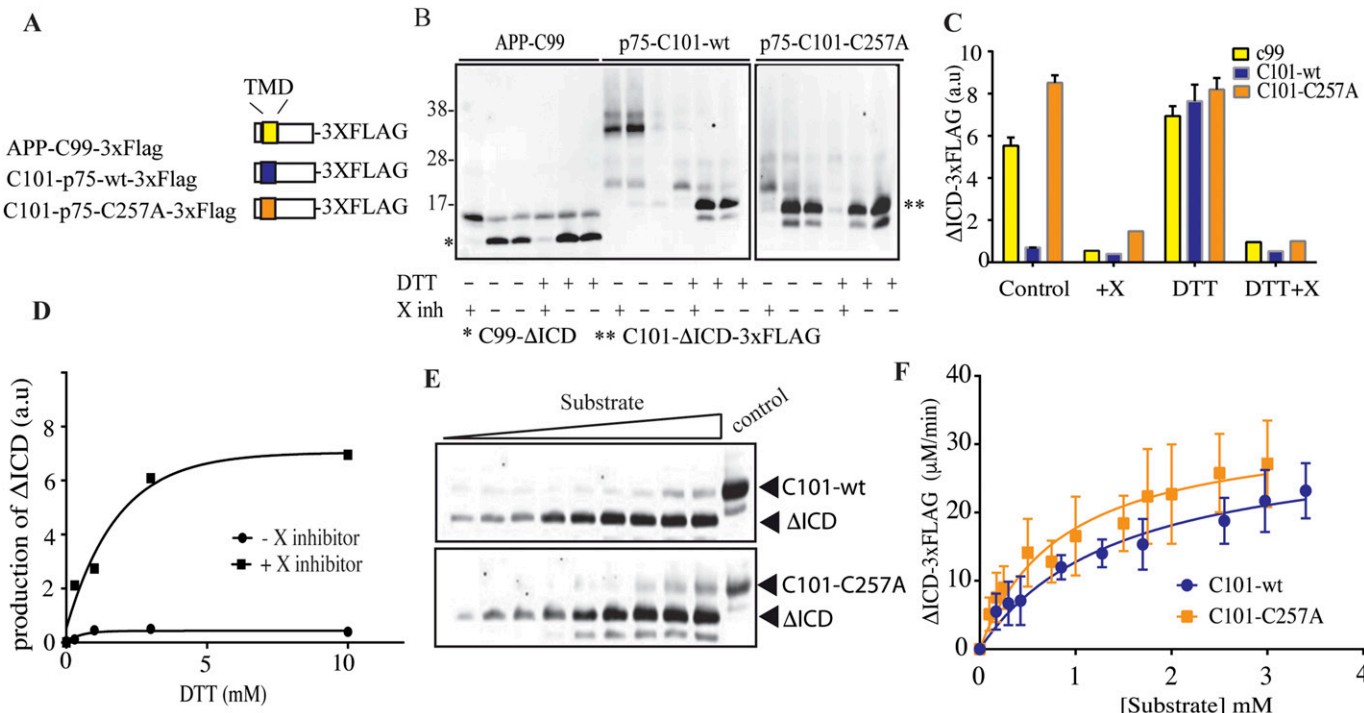

**Figure 3. p75 disulfide dimers are not processed by γ-secretase.**
**(A, B)** Schematic representation of the purified constructs (B). In vitro activity assays with purified human γ-secretase and APP-C99, p75-C101-wt, and p75-C101-C257A substrates incubated for 1 h in the presence or not of DTT reducing agent (20 mM) and γ-secretase inhibitor X (10 μM). (*) and (**) indicate C99-ΔICD and C101-ΔICD-3xFLAG product bands, respectively. **(C)** Total ΔICD-3xFLAG product levels analyzed by quantitative Western immunoblot reveal that only reduced p75-C101-wt substrates are processed by γ-secretase. **(D)** In vitro activity assays using purified human γ-secretase and p75-C101-wt over a DTT gradient. Notice that increasing DTT concentrations favor C101-wt processing. **(E)** Western blot of de novo ΔICD-3xFLAG generated from C101-wt and C101-C257A at 37°C upon incubation with the purified γ-secretase. **(F)** ΔICD-3xFLAG product generation was fit with a Michaelis–Menten model. Processing of C101 substrates into ΔICD (mean ± SEM) fit with Michaelis–Menten model (fit ± 95% CI) indicates similar values for C101-wt and C257A monomeric substrates. Kinetic parameters were obtained using the GraphPad Prism 6 software and are shown in Table 1. Bars represent the standard errors.

γ-secretase endoproteolytic cleavage of C101-p75-wt was increased in a DTT-concentration dependent manner, indicating that the reduction in the disulfide bond is necessary for γ-secretase endoproteolytic cleavage (Fig 3D).

To evaluate the catalytic efficiency of γ-secretase under equal kinetic conditions, we performed in vitro activity assays at 1 μM substrate concentration for C101-p75 and C101-p75-C257A previously DTT reduced substrates. Under these reduction conditions, the kinetic parameters of C101-p75-wt and C101-p75-C257A exhibit similar values, suggesting that in monomeric state, both wt and C257A mutant are equally processed by γ-secretase (Fig 3E and F and Table 1).

Altogether, our data indicate that covalent disulfide-linked p75-CTF dimers are resistant to γ-secretase processing, this feature results in the increased accumulation of dimeric forms and concomitant exacerbated induction of cell death.

## TrkA reduces p75-CTF-induced cell death

Studies with sensory and motor neurons have shown that during normal aging, there is a progressive increase in p75$^{NTR}$ expression that is accompanied by a parallel decrease in TrkA levels (Bergman et al, 1999; Johnson et al, 1999). The lowering in TrkA expression during aging, when considering the role of TrkA in mediating p75CTF internalization showed above, may be physiologically relevant in the neuronal death mediated by p75-CTF. To explore the role that TrkA plays in the prevention of p75-mediated cell death, we took advantage of PC12 cells and their variant PC12nnr5. These cells closely resemble sympathetic ganglion neurons but, as mentioned above, whereas PC12 cells express physiological levels of p75 and TrkA, the mutant PC12nnr5 variant does not express TrkA (Loeb et al, 1991). Cell treatment with the GSI CE for 72 h caused an increase in the percentage of PC12nnr5 apoptotic cells, as shown by caspase-3

absence or presence of TrkA plus NGF. Transfected cells were treated with NGF (50 ng) for the indicated times and immunostained for cell surface (Alexa 555) and intracellular (Alexa 488) p75-CTF. Higher magnification is shown at the 120-min time point. **(E)** Quantitative analysis of the confocal images of panel (D) shows a higher internalization of p75-CTF C257A mutant respect the wt. **(F)** Inhibition of p75-CTF internalization in the presence of the Trk inhibitor K-252a or the macropinocytosis inhibitor amiloride in HEK293 cells. **(G, H)** Representative confocal images of kinetic internalization experiments in PC12 and PC12nnr5 cells expressing wt or mutant HA-p75-CTF constructs in the absence or presence NGF. All the data are represented as mean ± SEM, N = 3. Two-way ANOVA followed by Tukey's post-test were used to determine the statistical significance, *P < 0.05, **P < 0.01, ***P < 0.001, ****P < 0.0001.

**Table 1. Kinetic parameters of p75-CTF and p75-CTF-C257A cleavage.**

| | P75-CTF | P75-CTF-C257A |
|---|---|---|
| Vmax (pM/min) | 31.26 ± 3.97 | 33.03 ± 4.22 |
| KM (mM) | 1.449 ± 0.43 | 0.864 ± 0.30 |
| 95% Confidence intervals | | |
| Vmax | 23.09–39.43 | 24.42–41.64 |
| KM | 0.5563–2.342 | 0.2566–1.473 |

cleavage immunofluorescence (Fig 4A and B). In contrast, PC12 cells did not exhibit any significant increase in cell death upon inhibition of γ-secretase. In these cell lines, p75[NTR] processing is physiologically regulated by RIP and accordingly, γ-secretase inhibition with CE induced p75CTF accumulation (Fig 4C). Interestingly, incubation with CE induced the formation of p75 oligomers that were cross-linked with the membrane impermeable cross-linker BS3, in PC12nnr5 but not in PC12 cells (Fig 4C). We also observed that apoptosis is accompanied by a significative increase in phosphorylated p38 levels in PC12nnr5 cells, an increment that is not seen in PC12 cells (Fig 4D and E).

We note that PC12 and PC12nnr5 cells are not necessarily the same cell line and other hidden mutations, further than the lack of TrkA, could affect the results. Therefore, we performed a rescue experiment by transiently re-expressing TrkA in PC12-nnr5 cells (Fig 4F and G). PC12-nnr5 cells were transfected either with TrkA+GFP or with just GFP as a control (Fig 4G) and subjected to GSI CE treatment for 72 h. Consistently with our previous results, γ-secretase inhibition significantly increased the percentage of cleaved caspase-3/GFP positive cells in the control PC12nnr5 cells (Fig 4H). However, TrkA re-expression rescued the cells from p75-mediated cell death upon γ-secretase inhibition (Fig 4H).

### TrkA activation disrupts p75-CTF oligomerization at specific plasma membrane domains

To explore the mechanism of p75CTF-mediated cell death and its inhibition by TrkA, we determined the oligomerization state of p75CTF at the plasma membrane. In vivo studies using the membrane-impermeable cross-linker BS3 showed that p75-CTF–transfected HeLa cells present dimers (ca 50 kD), tetramers (100 kD), and oligomers (>200 kD) in their plasma membrane, indicating cross-linking events between the lateral association of monomers and dimers (Fig 5A). To rule out that isolated p75-CTF over-expression may induce the formation of aberrant oligomers, HeLa cells were transfected with full-length p75[NTR] (p75-FL) and stimulated with PMA to induce the CTF generation. BS3 cross-linking experiments in these cells mimicked the results of p75-CTF over-expression (Fig 5B), indicating that oligomers formation at the plasma membrane also takes place when the p75-CTF levels are controlled by the endogenous α-secretase.

To evaluate TrkA contribution to p75-CTF oligomerization and toxicity, we quantified the p75-CTF oligomerization degree in HeLa cells co-transfected with TrkA. Enhanced stabilization of p75-CTF membrane oligomers by BS3 cross-linking in vivo showed that co-expression with TrkA significantly depleted p75-CTF dimers and multimers (two-way ANOVA analysis; construct factor $F_{(3, 6)}$ = 772 $P$ < 0.0001; BS3 treatment factor $F_{(1, 2)}$ = 954, $P$ = 0.0001; both factors $F_{(3, 6)}$ = 138 $P$ < 0.0001) (Fig 5C and D).

Protein oligomerization in membranes can be regulated by many factors, being one of them the membrane lipid composition. In this regard, several studies implicate cholesterol as a major player in protein oligomerization (Paladino et al, 2004; Ishitsuka & Kobayashi, 2007). Interestingly, p75-CTF has been localized to cholesterol-rich regions at the plasma membrane (Underwood et al, 2008), where γ-secretase activity concentrate (Matsumura et al, 2014). We identified a putative cholesterol recognition/interaction amino acid consensus sequence (CRAC) of the form $(L/V)-X_{1-5}-(Y)-X_{1-5}-(K/R)$ in the trans-membrane domain of p75[NTR] (Fig 5E). Of note, the tyrosine residue of CRAC motifs plays a key role in the interaction of cholesterol with different proteins (Fantini & Barrantes, 2013). Thus, we mutated the CRAC motif of p75[NTR] (to generate a mutant p75CTF (AxxAxxA)) to address its relevance on p75-CTF oligomerization (Fig 5E and F). The analyses showed that CRAC motif mutation in p75-CTF disrupts oligomer formation in the plasma membrane as showed by BS3 cross-linking experiments (Fig 5F).

As cholesterol-rich domains are locations where receptor endocytosis takes place and taking into account that TrkA reduces oligomer formation and mediates p75CTF internalization (Fig 2), we asked which pathways could be induced by TrkA to favor p75CTF internalization. It has been shown that phosphatidylinositol 4,5-bisphosphate (PIP₂) may play an important role in receptor internalization (Jost et al, 1998; Brown, 2015). We first wondered if modulation of the total levels of PIP2 plays any role in p75CTF oligomerization and internalization. Indeed, co-expression of p75CTF with synaptojanin, a PIP₂ phosphatase, reduced p75CTF oligomerization (Fig 5F) and increased p75CTF internalization (Fig S3A and B). As the activation of TrkA by NGF regulates the levels of PIP2 by the activation of the PI3K and PLCγ pathways (Soltoff et al, 1992), we further study the disruption of p75-CTF oligomerization by the use of different TrkA mutants (Fig 5G and H). Mutant TrkA receptors lacking the whole extracellular domain (TrkA-ΔECD) or just the immunoglobulin domains (TrkA-ΔIg), have been reported to render a constitutively active TrkA receptor (Arevalo et al, 2000), whereas on the other hand, deletion of the extracellular or intracellular juxtamembrane regions (TrkA-ΔeJTM or TrkA-ΔJTM, respectively), inactivate TrkA response to NGF. Co-expression of p75-CTF and TrkA mutants showed that constitutive activation of TrkA completely disrupted p75-CTF oligomerization. Notably, co-expression with inactive TrkA constructs (TrkA-ΔeJTM or TrkA-ΔJTM) did not affect p75-CTF multimerization (Fig 5H).

In agreement with these results, overexpression of the wt TrkA, that activates the kinase activity of TrkA (Fig 5I), or the constitutively active TrkA-ΔIg construct (Fig 5I) abrogated p75-CTF–mediated cell death upon γ-secretase inhibition (Figs 5J and S4), whereas the inactive TrkA-ΔeJTM form (Fig 5I) failed to rescue the apoptotic phenotype in these cells (two-way ANOVA analysis; construct factor $F_{(5, 10)}$ = 79 $P$ < 0.0001; CE treatment factor $F_{(1, 2)}$ = 100, $P$ = 0.0098; both factors $F_{(5, 10)}$ = 53 $P$ < 0.0001) (Figs 5J and S4). Interestingly, the CRAC domain is needed for the p75-CTF–mediated cell death (Fig 5J). In conclusion, TrkA activation by NGF reduces the levels of p75CTF oligomers at the plasma membrane, promotes p75CTF endocytosis, and inhibits cell death.

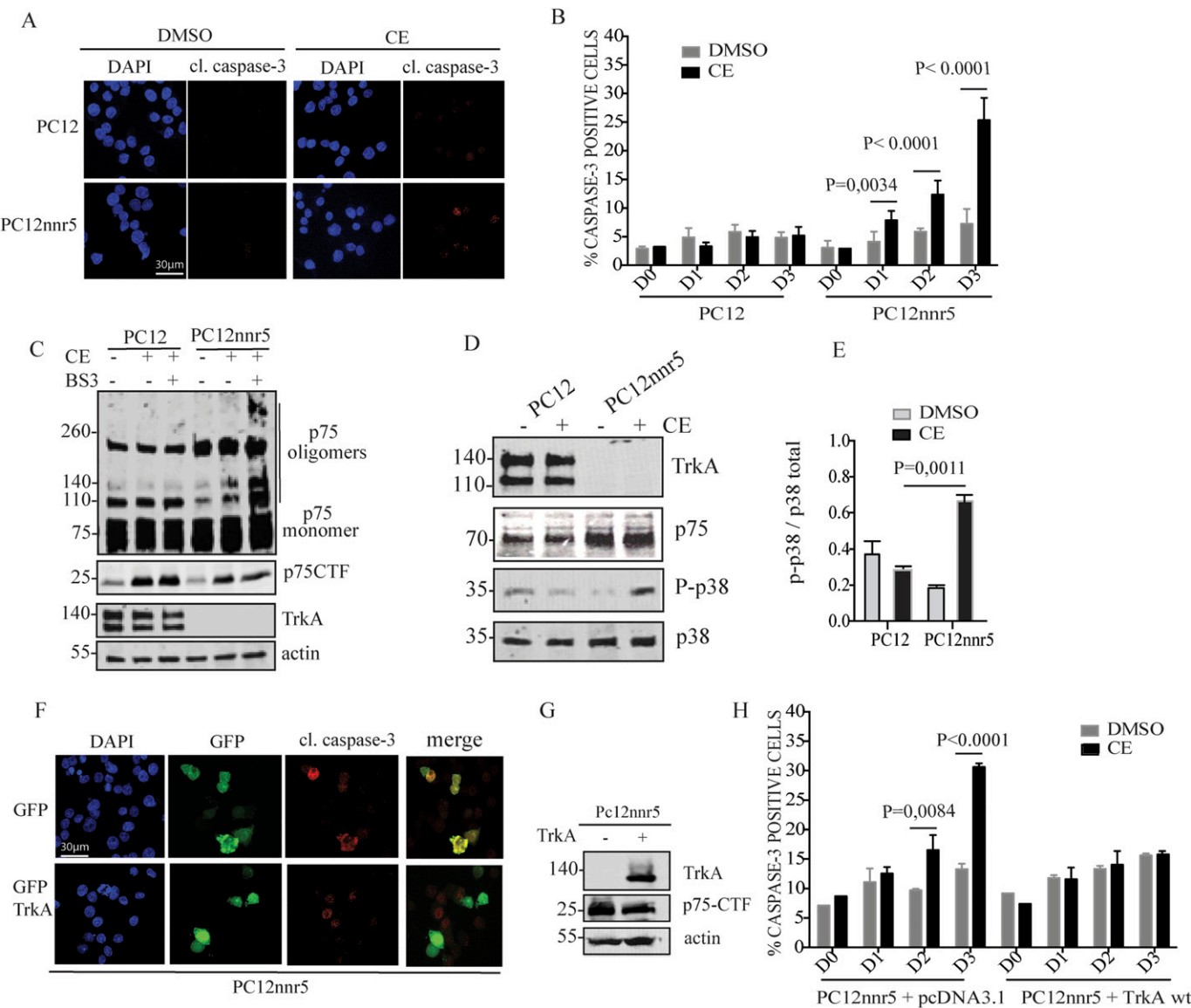

**Figure 4. TrkA reduces p75-CTF–induced cell death.**
**(A)** Confocal representative images of PC12 and PC12nnr5 cells incubated with CE (10 10 $\mu$M) over time and immunostained for DAPI (nuclei, blue) and cleaved caspase-3 (red). **(B)** Cell death quantification is represented as the percentage of cleaved caspase-3–positive PC12 cells incubated in presence or absence of CE for 1, 2, and 3 d. **(C)** Western blot of the lysates from the indicated cell lines treated with BS3 and CE. The levels of endogenous p75CTF and TrkA levels in PC12 and PC12nnr5 cells upon $\gamma$-secretase inhibition are shown. **(D)** $\gamma$-secretase inhibition induces p38 activation in PC12nnr5 cells. Representative SDS–PAGE/Western blot show endogenous levels of TrkA, p75[NTR] and P-p38 in PC12 and PC12nnr5 cells upon CE treatment for 72 h. **(E)** Densitometric analysis of the respective Western blot bands shows an increase in the ratio of P-p38/p38 signal after $\gamma$-secretase inhibition with CE in PC12nnr5 cells. **(F)** TrkA re-expression in PC12nnr5 cells rescues from cell death. PC12nnr5 cells transfected with either TrkA + GFP or with GFP + backbone vector (pcDNA3.1) were treated as previously described. Confocal images show DAPI and cleaved caspase-3 staining in cells incubated with CE for 3 d. **(G)** Re-expression levels of TrkA in transfected PC12nnr5 cells were confirmed by Western blot analysis. **(H)** Apoptotic cell death was quantified over time as described above. Cell death observed after CE treatment for 72 h is rescued upon re-expression of TrkA. More than 500 transfected PC12 cells were quantified per condition. All the data are represented as mean ± SEM, $N$ = 3; two-way ANOVA and Tukey's post-test was used to determine the statistical significance. $P$-values are showed in the graphics. *$P$ < 0.05, **$P$ < 0.01, ***$P$ < 0.001, ****$P$ < 0.0001.

## TrkA inhibits recruitment of TRAF6 to p75CTF oligomers

To further characterize the cell death mediated by p75-CTF, we focused our efforts on the molecular mechanism behind this signal transduction. JNK and p38 MAPK modulate cell programs for cell survival and differentiation and have been previously associated with p75-mediated caspase-3 activation (Harrington et al, 2002;

Jiang et al, 2005; Pham et al, 2016). Interestingly, we showed above that in PC12nnr5 cells, apoptosis is accompanied by a significant increase in phosphorylated p38 levels that is not observed in PC12 cells (Fig 4D and E) despite that both cell lines present similar p75-CTF levels (Fig 4C). Therefore, we assessed p38 and JNK phosphorylation levels in HeLa cells co-expressing p75-CTF and TrkA (Fig 6B and C) under GSI conditions. As shown by immunoblots, the

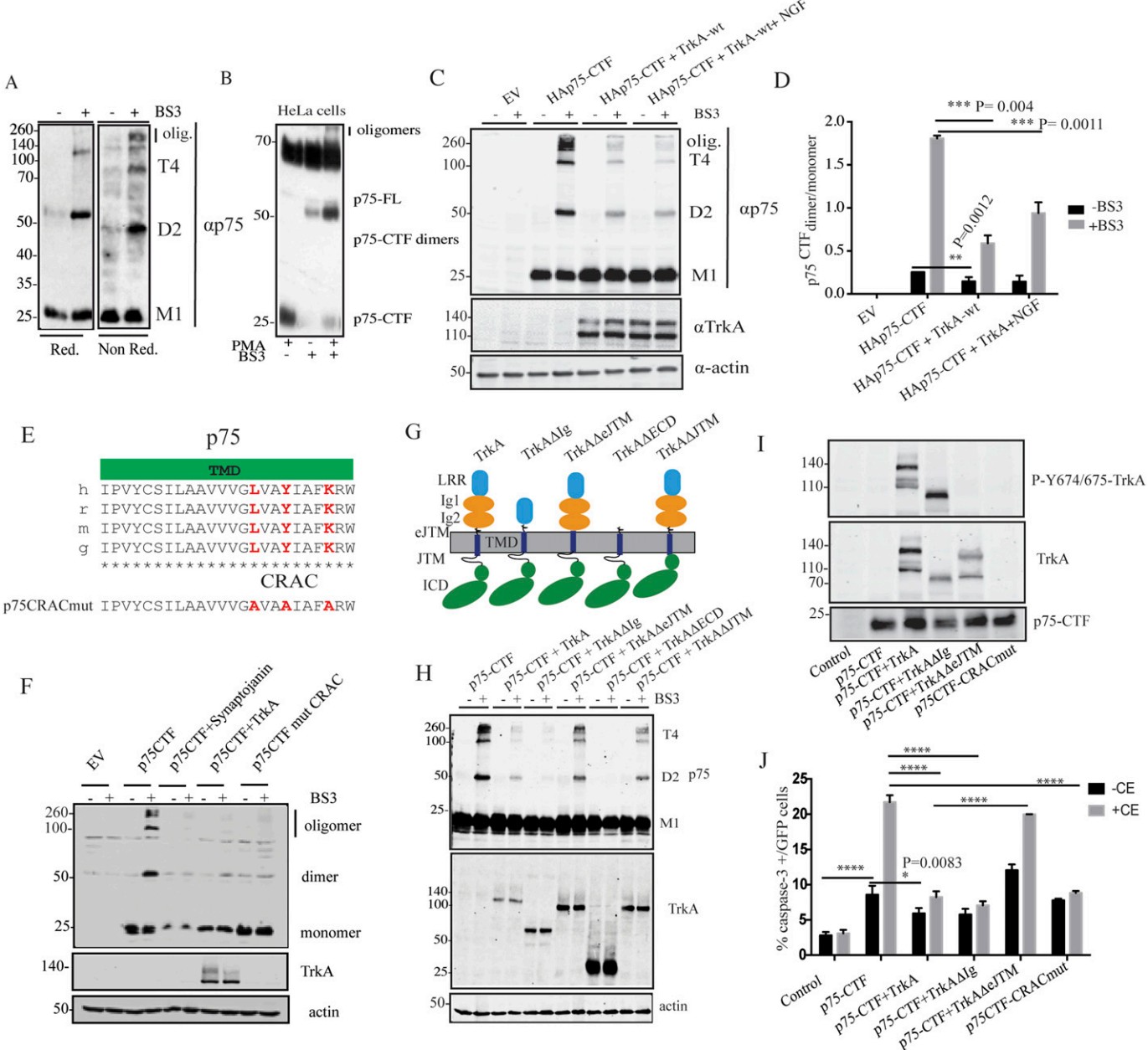

**Figure 5. p75-CTF oligomerization at the plasma membrane induces cell death.**
**(A)** Representative images of Western immunoblot analysis from p75-CTF–transfected HeLa cells cross-linked in vivo with BS3. The images show p75-CTF monomers (M1), dimers (D2), tetramers (T4), and oligomers (olig) migration detected with p75 antibody and analyzed in reducing (left) or nonreducing SDS–PAGE (right). **(B)** Analysis of in vivo BS3 cross-linking experiment in HeLa cells overexpressing full-length p75[NTR] and treated with PMA (200 nM) for 40 min. In agreement with (A), p75CTF generated from full-length receptor, forms dimers and oligomers at the plasma membrane that can be identified under nonreducing conditions. **(C, D)** Analysis of in vivo BS3 cross-linking experiments of HeLa cells overexpressing the indicated constructs. The effect of TrkA expression in p75-CTF oligomerization is quantified as the ratio of p75CTF dimer/monomer. NGF treatment does not show a significative difference in p75-CTF oligomerization with respect to TrkA expression alone. **(E)** Schematic representation of the p75 TM domain structure-sequence alignment across different species (h, human; m, mouse; r, rat; and g, chicken). In red are highlighted the residues forming part of the consensus CRAC sequence. Below, the protein sequence of the p75CTFCRACmut. **(F)** Western blot showing that synaptojanin and mutations in the CRAC motif sequence, as well as the presence of TrkA, reduce the formation of p75-CTF dimers and oligomers in the plasma membrane. **(G)** Graphic illustration of the different TrkA mutants used in this study. **(H)** BS3 cross-linking of membrane proteins in HeLa cells expressing the indicated constructs. Cell lysates were analyzed by SDS–PAGE Western blotting with p75 (top) and different TrkA antibodies (down). The presence of p75-CTF dimers (D2) and tetramers (T4) is indicated at the right. **(I)** Western blot of the lysates from HeLa cells transfected with the indicated construts reproved with the indicated antibodies. **(J)** Quantification of apoptotic cell death detected by immunofluorescence in HeLa cells transfected with the indicated constructs and incubated in the presence or absence of the γ-secretase inhibitor CE (10 μM) for 24 h. Analysis of GFP/cleaved caspase-3 positive cells supports the role of TrkA kinase activity in the inhibition of p75-CTF–mediated cell death. *P*-values (****$P <$ 0.0001) were determined for the average of three independent experiments and statistical analysis was performed using a two-way ANOVA using Tukey's post-test to correct for multiple comparisons. Bars represent standard error. *$P < 0.05$, **$P < 0.01$, ***$P < 0.001$, ****$P < 0.0001$.

TrkA activation inhibits p75-CTF cell death    Franco et al.    https://doi.org/10.26508/lsa.202000844    vol 4 | no 4 | e202000844    **9 of 20**

expression of p75-CTF produces an increment on P-p38 and P-JNK levels upon CE treatment (Fig 6C) that correlates with a significant increase in cleaved caspase-3–positive cells (Fig 6A). In agreement with our previous findings, TrkA co-expression rescued cells from p75-CTF–mediated cell death and blocked p38 and JNK activation in the cells subjected to γ-secretase inhibition (Fig 6B and C).

A major intracellular effector of p75 signaling is the TNFR-associated factor 6 (TRAF6) (Khursigara et al, 1999; Gentry et al, 2004; Kisiswa et al, 2018). It has been shown that this molecular adaptor binds to the intracellular juxtamembrane sequence of p75 and regulates the signal transduction of p75-induced cell death in a JNK-dependent manner (Yeiser et al, 2004; Geetha et al, 2005). To explore the relevance of CTF oligomers in TRAF6 interaction and cell

death signaling, the cells were co-transfected with TRAF6 and either wt p75-CTF or p75-CTF-C257A mutant (Fig 6D). Immunoprecipitation analysis revealed a remarkable specificity, with TRAF6 binding only to p75-CTF dimers. This finding resembles the mechanism previously observed in the full-length receptor (Vilar et al, 2009a, 2009b), but interestingly, TRAF6 and p75-CTF dimers interact in a constitutive manner that does not rely on neurotrophin binding. Consistently with our previous results, TrkA co-expression reduces the interaction between p75-CTF and TRAF6 (Fig 6E) by competing with TRAF6 for the binding to p75-CTF (Fig 6F). Hence, our data suggest that p75-CTF oligomers, generated here upon γ-secretase inhibition, induce apoptosis in a TRAF6-, JNK-, and p38-mediated pathway, with TrkA inhibiting the deadly process.

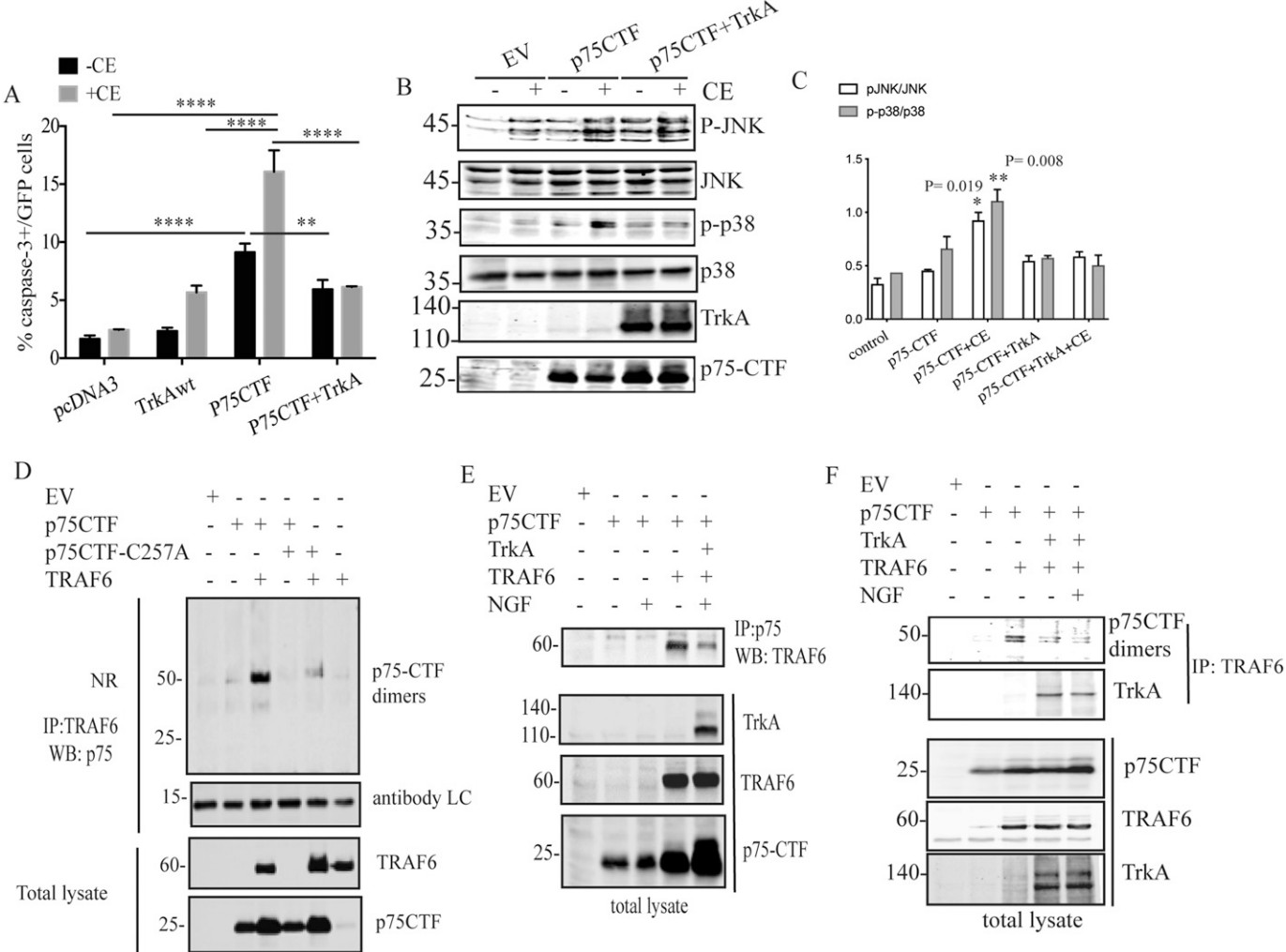

**Figure 6. p75-CTF recruits TRFA6 to activate JNK and p38 signaling cascades.**
**(A, B, C)** Inhibition of γ-secretase induces JNK and p38 activation in HeLa cells overexpressing p75-CTF. HeLa cells transiently expressing the indicated constructs were incubated in presence or absence of CE (10 μM) for 24 h. Apoptotic cell death was determined by immunofluorescence of caspase-3+/GFP+ cells and represented as mean ± SEM, N = 3; two-way ANOVA and Tukey's post-test was used to determine the statistical significance. Representative Western blot analysis of transfected HeLa cell lysates shows the expression levels of the indicated proteins. **(D)** TRAF6 binds to p75-CTF dimers. TRAF6 and p75-CTF interaction was determined by co-immunoprecipitation using anti-FLAG antibody (FLAG-TRAF6) and Western blot detection of p75. Nonreducing SDS–PAGE/Western blot analysis shows the co-elution of TRAF6 and p75-CTF dimers. TRAF6 and p75-CTF expression levels are showed in the total lysates. **(E, F)** Effect of TrkA on TRAF6 and p75-CTF interaction. Western immunoblots of the co-immunoprecipitation of TRAF6 with p75-CTF in the presence or absence of TrkA and NGF.
Source data are available for this figure.

### TrkA and γ-secretase inhibition induces BFCNs death in a p75-dependent manner

Although p75 is widely expressed during development, only some populations of neurons retain its expression in the adult CNS. These populations include the BFCNs, where p75[NTR] and TrkA are present in relatively high levels and regulate cell survival functions (Counts & Mufson, 2005). BFCNs participate in several cognitive processes by cortical and hippocampal innervation and consistently, their degeneration during normal aging and AD present substantial consequences for cognitive function (Boissière et al, 1996; Granholm et al, 2000; Mufson et al, 2008; Schliebs & Arendt, 2011; Koulousakis et al, 2019). To assess the effect of γ-secretase inhibition on BFCNs, we isolated BFCN from E17 embryonic mice and cultivated them for 11 days in vitro (11 DIV) to ensure their complete maturation and proper expression of choline acetyltransferase (ChAT). Mature cholinergic neurons were identified by ChAT, p75 and TrkA immunofluorescence (Fig 7A) and incubated with GSI Compound E for three consecutive days. Although inhibition of γ-secretase was no toxic for BFCN, the impairment of TrkA signaling produced some cell death that was consistent with reported in vivo data (Fagan et al, 1997). Strikingly, the combinatory treatment of CE and the specific TrkA inhibitor K-252a produced a major increase in the percentage of apoptotic cells after 3 d in culture (Fig 7B). Furthermore, cell death was completely rescued in BFCNs from p75-KO subjected to the same GSI and K-252a treatment, demonstrating that the event is p75-dependent (Fig 7C). Together, the data suggest that in adult neuronal populations with high expression levels of p75[NTR] and impairment of TrkA activity, as occurs in elderly BFCN, treatment with GSI drives to cholinergic neuronal apoptosis and cell death.

## Discussion

Regulated intramembrane Proteolytic processing of p75[NTR] underlies its apoptotic signaling, but the molecular mechanisms underlying its toxicity are not fully understood. Although p75[NTR] oligomerization is still a matter of debate (Lin et al, 2015; Chao, 2019; Goncharuk et al, 2020), several lines of evidence support the role of transmembrane dimerization for p75[NTR] biological activity (Tanaka et al, 2016; vilar et al, 2009a, 2009b). In agreement recent structural analysis reveal p75-TMD as a homodimer (Nadezhdin et al, 2016). However, p75[NTR] single-particle tracking in transfected cells recently determined the presence of p75[NTR] monomers at the plasma membrane (Marchetti et al, 2019). Of note, the N-terminal p75[NTR] tagging used in this analysis only provided insights into the oligomerization state of the full-length receptor and did not inform on the status of p75[NTR] after α-secretase shedding.

Here, we investigated the stoichiometry of p75-CTF and its role in the receptor-mediated cell death. Our in vitro cultures of DRG neurons showed that blocking the p75-CTF turnover by γ-secretase inhibition produced an accumulation of endogenous p75-CTF that leads to the enhanced formation of dimers. This evidence, together with the observation of p75-CTF disulfide dimers and oligomers in purified membranes from transfected p75-CTF cells, supports the hypothesis that p75-CTF oligomerizes under conditions that increase its concentration. And importantly, the formation of p75-CTF oligomers directly correlates with an increase in the cell death. Furthermore, although our cross-linking studies relay on a membrane impermeable reagent is still possible that oligomerization of p75CTF could also take place in the internalized membrane vesicles.

We cannot discard the possibility that the detected oligomers are adducts of p75[NTR] with other membrane proteins, as the presence of p75-CTF oligomers was detected by (non-selective) in vivo cross-linking assays. The data led us to propose that the increment of p75-CTF levels promotes the formation of oligomers by a mechanism that involves the oligomerization of the transmembrane domain. Although the participation of the death domain could play a role in the oligomerization of p75-CTF (Vilar et al, 2014; Lin et al, 2015), recent data contradict those results (Mineev et al, 2015; Goncharuk et al, 2020) and it will need further clarification. Interestingly, overexpression of p75-CTF-C257A also induced cell death, indicating that the covalent dimerization thorough Cys257 is not essential for the oligomerization and consequent p75-CTF–mediated apoptosis.

The localization of plasma membrane receptors in specific molecular compartments has been shown to play a relevant role in the cellular response. It has been reported that p75-CTF localizes into lipid rafts (Underwood & Coulson, 2008; Underwood et al, 2008) and cholesterol levels play a key role in p75-CTF pro-apoptotic function as its depletion abolish p75-CTF–mediated apoptosis (Underwood & Coulson, 2008; Underwood et al, 2008). We hypothesized that p75-CTF oligomers are stabilized by cholesterol and to challenge this hypothesis we disrupted the putative cholesterol-binding domain (CRAC domain) present in the transmembrane domain of p75[NTR]. Our analyses show that mutations on the p75[NTR] CRAC domain disrupt p75-CTF oligomerization and abolish the cell death effect observed upon p75-CTF turnover inhibition. Our results thus indicate that p75-CTF oligomers are the actual mediators of p75-mediated toxicity and indicate that their formation depends on its association with cholesterol through the p75[NTR] CRAC domain. This would explain the recent observations of Marchetti et al (2019), where cholesterol addition also confers apoptotic capability to the cysteine mutant p75[NTR] (Marchetti et al, 2019). Of note, cholesterol rich domains also play a role in the receptor internalization. It is known that in neurons p75[NTR] could be internalized through clathrin-dependent and clathrin-independent pathways depending of the presence of ligand neurotrophin, each one leading to different sorting pathways, like receptor recycling or axonal transport (Bronfman et al, 2003; Deinhardt et al, 2007). In this context, the finding that oligomerization of p75CTF is modulated by cholesterol could be related to the targeting of these oligomers to specific plasma membrane locations where internalization and the sequential sorting to different internalized endosomes would take place.

In this regard, we found that PIP$_2$ levels also contribute to the formation or stabilization of p75CTF oligomers. It has been shown that overexpression of p75CTF increases the activity of PIP 5-kinase, which is usually enriched in the plasma membrane, leading to more synthesis of PIP$_2$ (Coulson et al, 2008). PIP2 levels can also be regulated by the action of PTEN. It has been described that pro-NGF binding to p75 induces the expression and the activity of PTEN in basal forebrain neurons to counterbalance the pro-survival effects

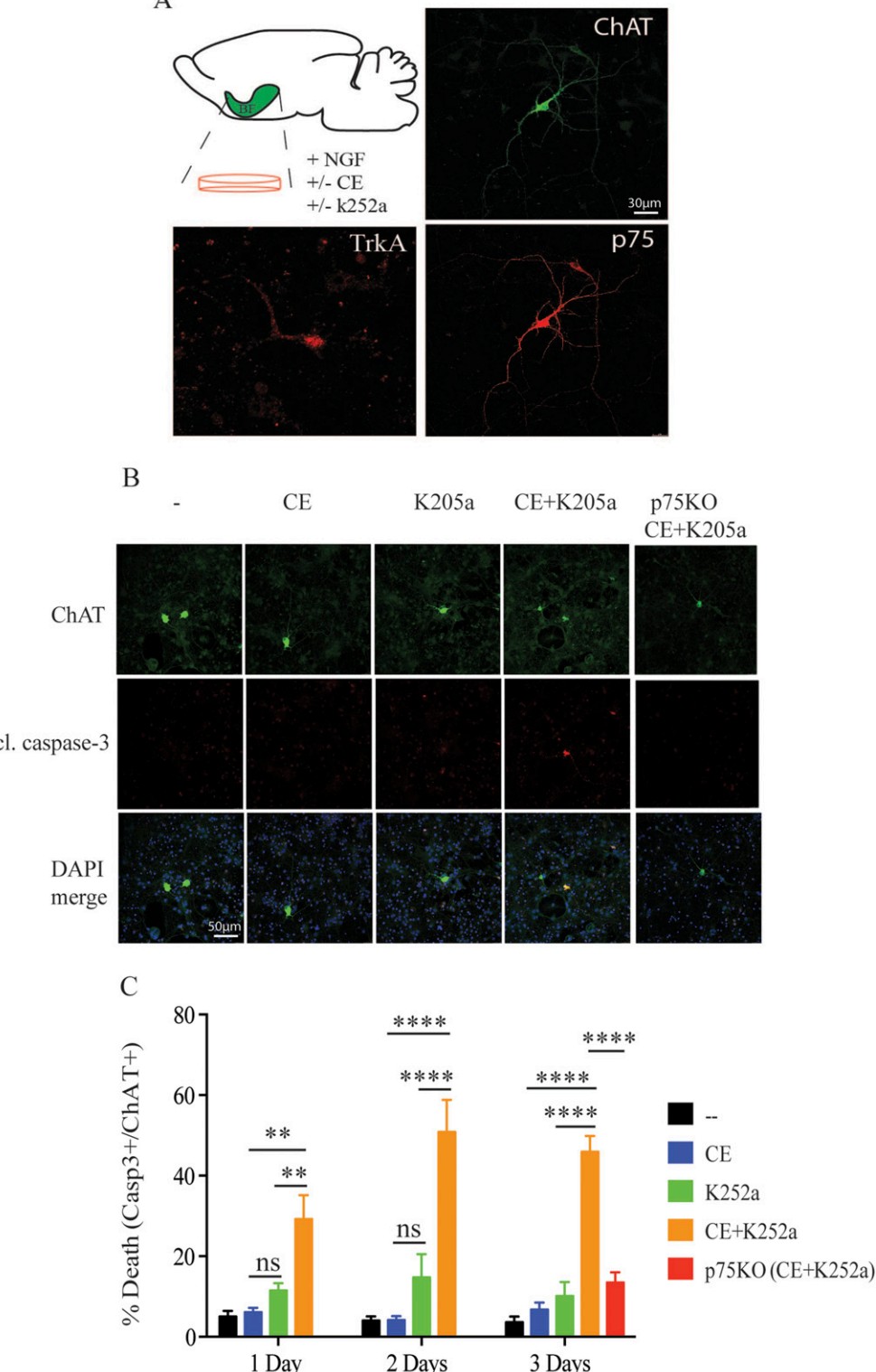

**Figure 7. TrkA and γ-secretase inhibition induces death of basal forebrain cholinergic neurons in a p75-dependent manner.**
**(A)** Graphical illustration of the basal forebrain area and the treatments used in these experiments (left). Representative confocal images from immunofluorescences of mature basal forebrain cholinergic neurons (BFCN) stained for ChAT, TrkA and p75[NTR] at DIV 11. **(B)** Representative confocal images from immunofluorescence of BFCNs stained for ChAT and cleaved caspase-3 upon the indicated conditions. **(C)** Apoptotic cell death analysis of mature BFCN (ChAT+) from wt and p75-KO mice at DIV 11. BFCN were incubated with DMSO, compound E, and TrkA inhibitor, k2025a, over 3 d and stained for cleaved caspase-3. Quantification of cleaved caspase-3+/ChAT+ cells in the respective immunofluorescences shows a significant cell death increase upon inhibition of TrkA and γ-secretase that is rescued in p75-KO BFCNs. All the data are represented as mean ± SEM, $N$ = 3. Two-way ANOVA followed by Tukey's post-test were used to determine the statistical significance. *$P$ < 0.05, **$P$ < 0.01, ***$P$ < 0.001, ****$P$ < 0.0001.

of TrkA (Song et al, 2010). The modulation of the levels of PIP$_2$ at the plasma membrane modulates receptor internalization (Brown, 2015). The decrease in PIP$_2$ by specific phosphatases such as synaptojanin or phospholipases, such as phospholipase C (PLC), is important to promote the pinch-off of the plasma membrane and the production of the endocytic vesicle (Cremona et al, 1999). TrkA activation by NGF modulates the levels of PIP$_2$ by activating the PI3K or the PLCγ pathways. Our results suggest that one of the

mechanisms of pro-survival roles of TrkA is to regulate the local levels of PIP2 around the oligomers of p75CTF and facilitate p75CTF internalization to specific endosomes. Our data showing that over-expression of synaptojanin reduces p75CTF oligomerization and increases p75CTF internalization support this hypothesis. We found that inhibition of macroendocytosis by TrkA inhibits p75CTF internalization. Macropinocytosis activation by TrkA has been previously shown to sort TrkA to signaling endosomes in neurons (Shao et al, 2002). Collectively, these findings suggest that in normal conditions some p75CTF could be sorted to these specialized signaling endosomes as it has been suggested by others (Bronfman, 2007; Urra et al, 2007).

Although internalization of p75CTF is slower in the absence of TrkA activation, it occurs at later time points (>120 min). The finding that cell death in PC12nnr5 cells and in cholinergic neurons with K-252a is significant only after 48–72 h suggested that cell death is a slow process that may require a constant accumulation of endosomes enriched in p75CTF oligomers. Clustering of TNFR1 or CD95, receptors from the TNFR superfamily such as p75$^{NTR}$, are known to induce its endocytosis and apoptosis signaling from the these internalized endosomes (Schütze & Schneider-Brachert, 2009). One property of p75CTF oligomers is that they are resistant to down-regulation by the γ-secretase. γ-Secretase activity mainly resides in cholesterol enriched lipid rafts of Golgi and endosome membranes (Vetrivel et al, 2004). We evaluated the capacity of γ-secretase to process monomeric versus dimeric p75-CTF substrates using in vitro activity assays. Remarkably, our results demonstrated that naturally occurring p75-CTF covalent dimers are resistant to γ-secretase cleavage. Moreover, our analysis reveals a correlation between substrate DTT reduction and cleavage product generation, indicating that the reduction in the transmembrane disulfide bond is required for γ-secretase cleavage. This finding is supported by the direct determination of Michaelis–Menten constants (Table 1) for wt and mutant p75-C257A reduced substrates. The impact that substrate homodimerization has on γ-secretase–mediated proteolysis has been a matter of controversy (Langosch et al, 2015; Winkler et al, 2015). Although, the generation of engineered APP-C99 substrates forming (covalent or no covalent) dimeric structures and their analysis by in vitro γ-secretase activity assays has shown that homodimerization protects the APP-C99 fragment from γ-secretase cleavage (Winkler et al, 2015). Our studies show for the first time that γ-secretase is not able to cleave a naturally dimeric p75-CTF substrate.

Based on these findings we propose a model (Fig 8) where the inhibition of the γ-secretase leads to an increase in the levels of p75-CTF which in turn that promotes its oligomerization in cholesterol/PIP2 rich regions at the plasma membrane. Recently Bronfman and collaborators showed that in sympathetic neurons p75$^{NTR}$ is internalized upon barin derived neurotrophic factor (BDNF) binding and directed to multivesicular bodies where it can be exocytosed in the form of exosomes (Escudero et al, 2014, 2019). It is highly possible that the p75CTF oligomers characterized here may follow a similar pathway, taking into account that BDNF does not activate TrkA and the recent report showing that the APP-CTF (C99) localizes to brain extracellular vesicles upon γ-secretase inhibition (Lauritzen et al, 2019). This suggests a general mechanism of CTFs disposal or, more interesting, the dispersal of a neurodegenerative signal.

Our data show that p75-CTF oligomers are constitutively bound to TRAF6 leading to JNK/p38 activation and cell death. The p75 juxtamembrane region contains a putative TRAF6-C recognition site (Vilar, 2017) and the dimeric nature of TRAF6 N-terminal region confers it a preferential binding for p75NTR dimers (Vilar et al, 2009b), whereas its C-terminal region, has a trimeric symmetry that could allow the formation of a high-molecular weight oligomers network (Yin et al, 2009). Thus, we propose that the newly formed p75-CTF dimers and oligomers recruit TRAF6 and the interaction trigger cell death through activation of the JNK/p38 signaling pathways. These findings are in agreement with a recent study reporting that pro-NGF binding to p75$^{NTR}$ induces TRAF6 recruitment and JNK activation, leading to cell death in cerebellar granule neurons (Kisiswa et al, 2018). Furthermore, it has been described that TRAF6 can be recruited to lipid rafts after activation of other TNFR receptors, like RANK by its ligand RANKL (Ha et al, 2003a, 2003b). In sympathetic neurons stimulation of BDNF causes cell death mediated by p75 in a JNK-dependent manner (Escudero et al, 2019). Recently, p75 has been found in a special apoptotic endosome transported along the axon of these neurons (Pathak et al, 2018). The identity of the proteome of such endosome is still unknown. Based on our findings it would be interesting to know if TRAF6, or other TRAF members known to interact with p75 (Ye et al, 1999), form part of this pro-apoptotic signaling endosome. The finding that p75$^{NTR}$ could be transported in Rab7-positive endosomes in the axons of motor neurons (Deinhardt et al, 2007) together with the data showing that TRAF6 co-localized to Rab7-positive endosomes in immune cells (Yan et al, 2020) suggests this could be the case.

We found that TrkA kinase activity abrogates p75-CTF oligomerization, promotes p75CTF internalization and inhibits cell death upon γ-secretase inhibition. Collectively, these observations assign a key role to TrkA in the regulation of p75$^{NTR}$ deadly function. In agreement, our studies in cholinergic neurons show that the inhibition of TrkA along with γ-secretase exacerbates the cell death effect mediated by GSIs in a p75NTR-dependent manner. Of relevance, cholinergic neurons are one of the few populations of the CNS that express relatively high levels of p75$^{NTR}$ during adulthood and their severe lost during AD correlates with changes in hippocampal synaptic transmission and progression of dementia (Sze et al, 1997). When NGF activates TrkA it mediates p75 shedding and the internalization of p75CTF (Urra et al, 2007) probably by inducing macropinocytosis to sort TrkA/p75CTF to signaling endosomes (Shao et al, 2002; Valdez et al, 2005; Philippidou et al, 2011) with a survival and cholinergic differentiation role. The mechanisms described here may play key roles in specific pathological situations. In AD there is an imbalance between pro-NGF and NGF (Fahnestock et al, 2001, 2004; Pedraza et al, 2005). Binding of pro-NGF induces the shedding of p75 and the activation of PTEN (Song et al, 2010), increasing the levels of PIP$_2$ from the pro-survival role of PIP$_3$, which might deregulate p75 endocytosis. As Pro-NGF does not activate TrkA the activation of PI3K would be low. If in addition the activity of the γ-secretase is compromised by familiar mutations (Chávez-Gutiérrez et al, 2012) or by the use of GSIs, cell death events are exacerbated. Thus, our results may also acquire particular significance in the context the failed phase III clinical trial with the GSI semagacestat, where the unexpected cognitive decline of the

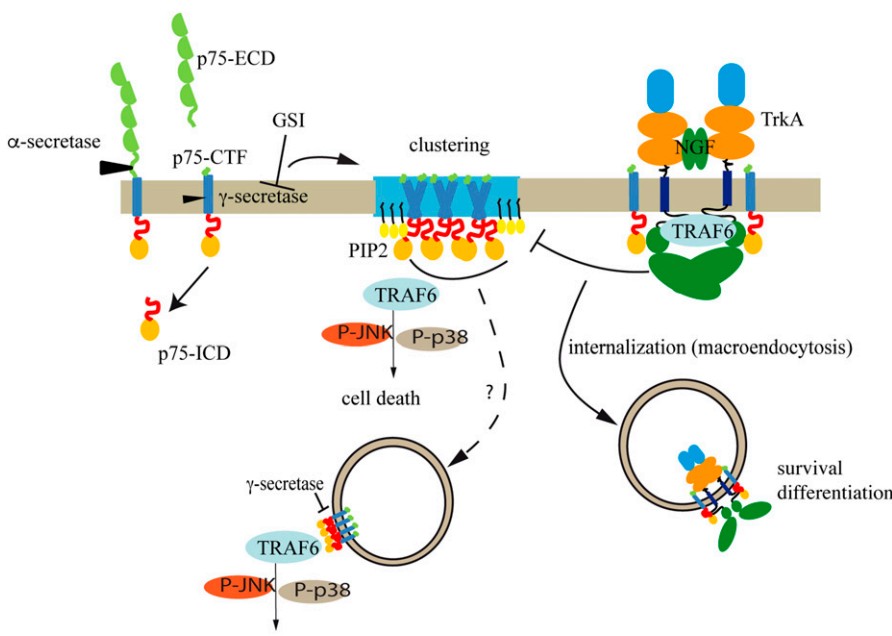

**Figure 8. Model of the results presented.**
Upon γ-secretase dysfunction, p75-CTF dimer and oligomerization in the cholesterol-rich region of the plasma membrane results in an increase of the PIP2 levels. Oligomers of p75CTF induce the activation of caspase-3 cleavage and cell death in a mechanism dependent of TRAF6, JNK, and p38. Alternatively p75CTF oligomers may be internalized and signal cell death from internalized vesicles (dashed line). TrkA kinase activity inhibits p75-CTF clustering and protects from cell death in part by decreasing the levels of PIP2 and promoting p75CTF internalization. Our data suggest that in a scenario where γ-secretase is inhibited, the final outcome would depend on the relative expression levels of p75 and TrkA in the cells.

treated group was observed (Doody et al, 2013). We speculate that the worsening in cognition observed in the semagacestat trial could be linked to the inhibition of p75-CTF turnover and its consequent accumulation in the cholinergic neurons of the treated AD patients. Of note, TrkA levels, but not p75NTR, are reduced in elderly AD patients (Mufson et al, 1996, 2000, 2002, 2008; Counts et al, 2004; Ginsberg et al, 2006). In vivo evaluation of the pathophysiological role of p75-CTF oligomerization warrants future research.

# Materials and Methods

### Cell lines culture

HeLa cells were cultured in DMEM (Gibco) containing 10% fetal calf serum (Thermo Fisher Scientific). PC12 cells were cultured in DMEM with 10% FBS and 5% horse serum. All cell lines were cultured at 37°C in a humidified atmosphere with 5% of $CO_2$.

### Antibodies

The following antibodies were used in immunoblotting and immunofluorescence experiments: rabbit anti-human p75NTR (1:1,000; G3231; Promega), rabbit anti-TrkA (1:1,000; Millipore), rabbit anti-phosphoTyr674/5 (1:1,000; Cell Signaling), mouse anti-HA (1:2,000; Sigma-Aldrich), mouse anti-FLAG M2 (1:1,000; Sigma-Aldrich), mouse anti β-actin (1:1,000; Sigma-Aldrich), rabbit MBP-probe (1:1,000; Santa Cruz), rabbit anti-Cleaved Caspase-3 (1:1,000, 9661S; Cell Signaling), rabbit anti phospho-p38 (1:1,000, 9211; Cell Signaling), rabbit anti p38 (1:1,000, 9212; Cell Signaling), rabbit anti JNK (1:1,000, 9252; Cell Signaling), rabbit anti phospho-JNK (1:1,000, 9251; Cell Signaling), goat anti-choline acetyltransferase (1:200, AB144P; Millipore), rabbit anti Cy3 (1:500; Jackson), goat anti mouse Ig/HRP

(1:10,000; Jackson), goat anti rabbit Ig/HRP (1:10,000; Jackson), goat IRDye800 (1:15,000; Rockland), and goat anti-mouse antibodies coupled to either Alexa 555 or Alexa 488 (Invitrogen). The DNA was stained with DAPI (1:1,000).

### DNA constructs design

p75NTR was expressed from the pcDNA3 vector backbone (Invitrogen) using a full-length coding sequence flanked by an N-terminal HA epitope tag. Mutations in C257A were introduced by direct mutagenesis using *Pfu* Turbo DNA polymerase (Agilent), and the oligonucleotide sequences are available upon request. p75-CTF contains the p75NTR signal peptide, an HA tag and the residues $R_{245}GTTDN_{250}$ from the p75NTR juxtamembrane region followed by the transmembrane domain and the intracellular region (see scheme in Fig 3). P75-CTF-C257A was made by direct mutagenesis from wt p75-CT. For expression and purification, the pSG5-C101-3xFLAG and pSG5-C101-C257A-3xFLAg were built on the vector pSG5-APPC99-3xFLAg by digestion with EcoRI and BamHI to eliminate the APP-C99 insert and ligation of the p75NTR insert produced from PCR amplification (the sequence of rat p75NTR inserted into the vector pSG5 is $M_{231}$VTTVMGSSQPVVTRGTTDNLIPVYCSILAAVVVGLVAYIAFKRWNSCKQNKQGANSRPVNQTPPPEGEKLHSDSGISVDSQSLHDQQTHTQTASGQALKG$_{332}$-3xFLAG, underlined in the transmembrane domain). TrkA point mutations and deletion constructs were built from pCDNA.3.1-HA-TrkA (a gift from Y Barde) using site-directed mutagenesis. DNA primers sequences will be distributed upon request. Synaptojanin2-pmCherryC1 was a gift from Christien Merrifield (plasmid # 27677; Addgene; http://n2t.net/addgene:27677; RRID:Addgene_27677) (Taylor et al, 2011).

### Isolation and primary culture of mouse DRG

E16-E17 mice were sacrificed, first spinal column was isolated, the head was removed by cutting at the base of the skull (C1-C2

level). The ribs were then cut parallel with and close to the spinal column on both sides, detaching the viscera connected to the anterior side of the spinal column; muscle, fat, and skin were cut from the posterior side of the spinal column using curved scissors, and the whole spinal column was put in a Petri dish containing 4°C Hank's balanced salt solution. The spinal cord was slowly peeled in a rostral to caudal direction from the column, revealing the DRGs below that were carefully removed so as not to damage them with the scissors. Three isolated DRG was collected on each cover slides of 24-well plates pre-coated with poly-D-lysine solution (100 µg/ml). DRG were cultured with DMEM containing 10% FBS, 1% glutamine, 0.5% penicillin–streptomycin, and 50 ng/ml NGF and were cultured at 37°C in humidified atmosphere with 5% of $CO_2$.

### Cell death quantification

Primary cortical neurons and Hela and PC12 cell lines were transfected with the indicated constructs (1 µg per 10 cm plate) and GFP in a 1:10 ratio respect to the main construct. 24 h after transfections, cells were lifted, counted and re-plated in 24 well plates. Cells were incubated for 24 h in presence or absence of CE (10 µM) before fixation. Washed cells were fixated with 4% PFA/PBS solution for 15 min at room temperature and permeabilized for 1 h with 0.1% Triton/PBS before staining for cleaved caspase-3. Cell death was analyzed by immunofluorescence and quantified as the percentage of GFP and cleaved caspase-3 double positive cells, respect to all GFP-positive cells.

### Membrane purification

HeLa cells were transiently transfected with wt p75-CTF or mutant p75-CTF-C257A expression vectors and collected 48 h post-transfections. Before collection, the cultures were incubated overnight (16 h) in presence or absence of GSI compound E 10 µM, to prevent p75CTF degradation. The plates not treated with compound E were incubated with DMSO as a control. Cells were collected and resuspended in 25-mm PIPES (pH 7), 120 mM KCl, 250 mM sucrose, 5 mM EGTA, and 1× complete protease inhibitor (Roche). Cell membranes were broken by mechanical processes and cell debris were removed by low centrifugation at 4°C. Total membranes were obtained after supernatant ultracentrifugation at 100,000$g$ for 1 h at 4°C. Pellet was resuspended in the same buffer previously described before cleavage experiments and incubated in the presence or absence of compound E for 1 h at 37°C.

### Cycloheximide treatment

HeLa cells were transfected with 1 µg of empty vector (control) or the indicated p75-CTF constructs. 48 h posttransfection, the cells were incubated in a six-well plate with 5 µg/ml cycloheximide (CHX; Sigma-Aldrich) in the presence and absence of 10 µM epoxomicin (Sigma-Aldrich), 1 µM wortmannin (Sigma-Aldrich), 20 mM ammonium chloride (Sigma-Aldrich), 50 nM Semagacestat (Selleckchem), and 10 µM compound E (Callbiochem). Cells were harvested in TNE lysis buffer (50 mm Tris–HCl, pH 7.5, 150 mM NaCl, 1 mm EDTA, 0.1% SDS, 0.1% Triton X-100, 1 mm PMSF, 10 mm NaF, 1 mm $Na_2VO_3$, 10 mm iodoacetamide, and protease inhibitor mixture), at different time

points (0, 1, 4, and 9 h) after CHX treatment. The half-lives were calculated using by densitometry of Western blots bands using the ImageQuant (Molecular Dynamics) software. Values were fit to the half-life decay equation using the GraphPad Prism software to an exponential regression of the form: $N_{(t)} = N_{(0\ h)} * e^{-\lambda t}$. $\lambda$ is the decay constant. Half-lives ($t_{1/2}$) were calculated using the equation $t_{1/2} = \ln(2)/\lambda$.

### Reducing and nonreducing SDS–PAGE

Protein lysates were analyzed using reducing or nonreducing SDS–PAGE. In reducing gels, sample buffer contains 5% of $\beta$-mercaptoethanol. We observed that in nonreducing conditions, p75-CTF samples run with a high smearing background and low levels of monomer are observed probably by the formation of high molecular weight aggregates. We found that the inclusion of a small amount of $\beta$-mercaptoethanol, 1%, eliminates the smear but retains the disulfide dimers mediated by the C257.

### p75 cleavage experiments

p75[NTR] cleavage experiments were carried out according to the protocol described previously by Kanning et al (2003). PC12 cells were incubated with NGF (100 ng/ml). 48 h after differentiation, PC12 cells were incubated for 90 min with either 1 µM proteasome inhibitor epoxomicin (Sigma-Aldrich), 10 µM Compound E (Millipore), or PBS buffer. Next, 200 nM phorbol 12-myristate 13-acetate (PMA; Sigma-Aldrich) was added for 40 min. Cells were washed in PBS and lysed in cold lysis buffer (50 mm Tris–HCl, pH 7.5, 150 mm NaCl, 1 mm EDTA, 0.1% SDS, 0.1% Triton X-100, 1 mm PMSF, 10 mm NaF, 1 mm $Na_2VO_3$, 10 mm iodoacetamide, and protease inhibitor mixture) at 4°C. Cellular debris was removed by centrifugation at 13,000$g$ for 15 min and protein quantification was performed by Bradford assay. Proteins were resolved by SDS–PAGE and membranes were incubated overnight at 4°C with rabbit polyclonal anti-human p75[NTR]. After incubation with the appropriate secondary antibody, membranes were imaged using enhanced chemiluminescence and autoradiography.

### Purification of γ-secretase

The purification of γ--secretase was carried out after a previous protocol (Acx et al, 2014). Briefly, HI5 insect cells were infected with baculovirus encoding human PSEN1, NCT-GFP, APH1A$_L$, and PEN-2. The GFP was cloned at the C-terminal site of NCT. γ-Secretase complexes were purified using agarose beads (NHS-activated beads; GE Healthcare) coupled with anti-GFP nanobodies. A PreScission cleavage site was included between NCT and GFP and used to elute untagged γ-secretase complexes. Removal of the GST-tagged PreScission protease was carried out by immunoaffinity pulldown using Glutathione Sepharose 4B (GE Healthcare).

### Recombinant protein production and extraction

COS1 cells were transiently transfected with wt pSG5-C101-p75wt-3xFLAG or mutant pSG5-C101-p75C257A-3XFLAG vector using using TransIT-LT1 (Mirus) according to the manufacture protocol. Before collection, cells were treated overnight with 10 µM Inhibitor X (Sigma-Aldrich) to prevent cleavage. Harvested cells were collected

by low velocity spin, resuspended in 50 mM Tris–HCl (pH 7.6), 150 mM NaCl, 1% Nonidet P-40, and complete protease inhibitor mixture (Roche) and incubated on ice for 1 h. Supernatant was obtained by ultracentrifugation at 100,000$g$ for 20 min. Immunoaffinity purification was carried out with the anti-FLAG M2-agarose beads (Sigma-Aldrich), according to the manufacturer's protocol. C101-p75-3xFLAG was eluted in 100 mM glycine HCl (pH 2.4), 0.625% n-dodecyl $\beta$-D-maltoside (Sigma-Aldrich) and immediately neutralized to pH 7 by the addition of Tris–HCl (pH 8.0).

### In vitro γ-secretase assay

In vitro activity assay was performed as previously described (Acx et al, 2014) with minor modifications. Purified γ-secretase (~15 nM final in assay) was incubated with purified C101-p75-3xFLAG or C99-3xFLAG at the indicated concentrations for 1 h at 37°C (in 15 $\mu$l final volume) were carried out in 25 mM PIPES (pH 7.0), 150 mN NaCl, 0.5% phosphatidylcholine, 0.25% CHAPSO, 2.5% DMSO, and 1 X EDTA-free complete proteinase inhibitors (Roche) at 37°C.

### Quantification of in vitro γ-secretase–mediated processing of p75

Previous to SDS–PAGE analysis, lipids and remaining substrate are extracted with chloroform/methanol (2:1, vol/vol). This extraction allows a better visualization of the reaction product and a more accurate measurement of the cleavage reaction efficiency. This process was carried out as previously described (Acx et al, 2014). Because of the extraction is not complete, the remaining substrate (indicated in a general form as p75-C101-3xFLAG both for wt or C257A) can still be present in diverse lanes. The amount of substrate extracted before gel loading cannot be controlled, so in this experiment, any change in p75-C101-wt monomer band corresponds only to a different extraction efficiency and does not affect the quantification of the C-terminal product. The C-terminal fragment-x3FLAG levels were determined by semi-quantitative Western blot using the anti-FLAG M2 antibody from Sigma-Aldrich and IR detection at 800 nm using the Odyssey Infrared Imaging System.

### Cell transfection

HeLa cells, which do not express endogenous p75 nor TrkA, were cultured in DMEM medium (Thermo Fisher Scientific) supplemented with 10% FBS (Thermo Fisher Scientific) at 37°C in a humidified atmosphere with 5% CO$_2$. Transfection of HeLa cells was performed using polyethylenimine (PEI; Sigma-Aldrich) at a concentration of 1–2 $\mu$g/$\mu$l. 48 h after transfection, the cells were starved in serum-free medium for 2 h, washed with PBS and incubated with BS3 in PBS for 15 min on ice. Cells were lysed with TNE buffer (Tris–HCl, pH 7.5, 150 mM NaCl, and 1 mM EDTA) supplemented with 1% Triton X-100 (Sigma-Aldrich), protease inhibitors (Roche), 1 mM PMSF (Sigma-Aldrich), 1 mM sodium orthovanadate (Sigma-Aldrich), and 1 mM sodium fluoride (Sigma-Aldrich). In the experiments involving the TrkA cysteine mutants, 10 mM iodoacetamide (Sigma-Aldrich) was added to the lysis buffer. Lysates were kept on ice for 10 min and centrifuged at 12,000$g$ for 15 min in a tabletop centrifuge. The protein level of the lysates was quantified using a Bradford kit (Pierce) and lysates were analyzed by SDS–PAGE.

### p75-CTF kinetic internalization assay

Hek293 cells were grown and transfected on sterile coverslips. The p75CTF expressed on the cell surface was labeled with the primary antibody (mouse anti-HA 12CA5, dilution 1:100) diluted in PBS for 1 h at 4°C and returned to the incubator at 37°C. For kinetic internalization experiments at different time points (0, 15, 30, 60, 120 min) the cells were fixed with 4% paraformaldehyde for 10 min at room temperature. Fixed cells were incubated with blocking buffer (0.1 M PB 3% FBS) for 45 min at room temperature with the Alexa 555 conjugate secondary anti-mouse Ig (Invitrogen). This was followed by a second incubation of blocking buffer containing 1% Triton X-100 (Sigma-Aldrich) to permeabilize the cells and a final incubation with the Alexa 488 anti-mouse secondary conjugate Ig (Invitrogen) for 45 min at room temperature. Receptor expression levels were determined by measuring the p75CTF fluorescence intensity at 561 nm (red) and 488 nm (green) light. Images of the cells were taken in a Leica SP8 spectral confocal microscope using a 63× magnification (oil).

### p75-CTF BS3 cross-linking

Transfected cells with the p75-CTF construct were washed three times with ice-cold PBS (pH 8.0), chilled on ice, and incubated in BS3 (bis[sulfosuccinimidyl] suberate) solution to a final concentration of 1 mM dissolved in PBS for 30 min at room temperature to cross-linker the membrane proteins. Free BS3 was quenched with 15 mM Tris, pH 7.5 for 15 min at room temperature. Then, the cells were washed twice with ice-cold PBS and lysed with TNE buffer (Tris–HCl, pH 7.5, 150 mM NaCl, and 1 mM EDTA) supplemented with Triton X-100 (Sigma-Aldrich) and a mixture of protease inhibitors (Roche Applied Science) and phosphatase-like sodium orthovanadate, Na$_3$VO$_4$ (Sigma-Aldrich), and sodium fluoride, NaF (Sigma-Aldrich). Proteins were subjected to SDS–PAGE and immunoblotted with the p75 intracellular antibody (dilution 1:10,000; Promega) to detect p75CTF.

### Western blot analysis

Cellular debris was removed by centrifugation at 12,000$g$ for 15 min and the protein level of cell lysates was quantified using the Bradford assay (Pierce). Proteins were resolved in SDS–PAGE gels and transferred to nitrocellulose membranes that were incubated overnight at 4°C with the indicated antibodies. After incubation with the appropriate secondary antibody, the membranes were imaged and bands quantified using enhanced chemiluminescence and autoradiography.

### Co-immunoprecipitation assay (co-IP)

A p100 plate of HEK293 cells were transfected with 5 $\mu$g of indicated plasmids. At 48 h post-transfection, cells were washed twice with ice-cool PBS and were lysed using 400 $\mu$l TNE buffer supplemented with 1% Triton X-100 and a mixture of protease inhibitors, orthovanadate, and sodium fluoride. Cells were harvested by scraping and transferred into a 1.5-ml tube, insoluble debris was removed by centrifugation (12,000$g$ for 10 min), and 100 $\mu$l of sample was reserved for input analysis.

The samples were incubated with indicated 2 µg of primary antibody (anti-Flag M2, anti-HA) overnight at 4°C with rotation and then incubated with 15 µl of Protein G Agarose resin 4 Rapid Run (4RRPG-5; Agarose Bead Technologies) for 2 h at 4°C with rotation. The beads were separated by gently centrifugation (2,800g for 1 min) and washed three times in 500 µl TNE buffer 0.5% Triton X-100. Finally, for nonreducing SDS–PAGE analysis, 30 µl nonreducing 2× sample buffer was added and the samples were boiled for 5 min at 96°C. For reducing SDS–PAGE analysis, 30 µl 2× sample buffer (with 2% β-MeOH) was added. For immunodetection, the indicated antibodies were used.

### Internalization of p75CTF by flow cytometry

HeLa cells were transfected with HAp75CTF and TrkA constructs. 48 h after transfection, the cells were incubated in the presence or absence of NGF for 60 min. Then the cells were lifted, washed with PBS, and counted. $10^6$ cells were resuspended in 200 µl of anti-HA primary antibody (1: 100 in PBS, the cytometry buffer), cells were incubated in suspension at 4°C for 30 min, with brief shaking every 10 min during incubation. After incubation, three washes with cold PBS were made by centrifugation for 5 min at 100g. A second incubation was performed with 100 µl of Alexa 488 anti-mouse secondary antibody (1: 100, in PBS) at 4°C for 30 min in the dark. Finally, the cells were washed two times with cold PBS and resuspended in 3 ml of PBS. The cell suspension was transferred to cytometry tubes and kept on ice to be analyzed by the Cytometer FACSCantoI (BD Biosciences) and analyzed with DiVa8 software.

### Isolation, culture, and transfection of embryonic cortical neurons

Late embryonic stage (E16-17) mouse fetuses were used. Fetuses were individually removed from the embryonic sac and placed into a sterile Petri dish. Mouse fetuses were decapitated, and the skin and skull were removed and the brain was place onto another Petri dish with cold HBSS. The cerebellum, olfactory bulb, meninges, and the non-cortical structures were carefully removed. Cortical hemispheres were cut into small pieces and transferred into a 15 ml conical tube. The cortical tissue was digested in 1 ml of 2.5 mg/ml of trypsin and 0.5 ml of 200 U/ml of DNaseI. The supernatant was replaced with 0.5 ml of 4% BSA and 1 ml of NB/B-27 and dissociated by passes through pipettes with decrease bore sizes. Cell suspension was centrifuged at 200g for 5 min. The pellet was suspended in 5 ml of 0.2% BSA and passed through 40 µm nylon filter, and the viable cells were counted. The cells were transfected in suspension with the AMAXA 4D-NUCLEOFECTOR using Amaxa P3 Primary Cell kit. $5 × 10^5$ cells were incubated with 1 µg of the indicated DNA plasmids, 17 µl of P3 solution and 3 µl supplement. The mixed was put in a well of a strip and DC100 AMAXA program were used. After transfection, the cells were suspended in 1 ml of plating medium (NB medium with B-27 supplement and 2 mM L-glutamine and 0.5% penicillin–streptomycin) and 500 µl were seeded onto two cover slides of 24-well plates pre-coated with 100 µg/ml poly-D-lysine and 5 mg/ml laminin. Next day, 500 µl of fresh NB, B27, and 2 µM AraC were added. Neurons were allowed to adhere and recover 4 d before the assays.

### BFCNs culture

BFCNs isolation protocol was adapted from Schnitzler et al (2008). Embryos of CD1 mice of 17–18 d were surgically removed and septo-hippocampal areas were dissected from the cerebral tissue in ice-cold Hanks balanced salt solution (HBSS, Gibco, Life Technologies), digested with 1 ml of trypsin, and 0.5 ml of 100 kU DNasa I (GE Healthcare) during 10 min at 37°C. The fragments were dissociated by aspiration with progressive narrower tips in 0.5 ml BSA 4% and 1 ml of Neurobasal medium (Gibco, Life Technologies) supplemented with 2% B-27 (Gibco, Life Technologies). After tissue disaggregation, 2.5 ml of BSA 4% was added, and the tubes were centrifuged for 5 min at 250g. The supernatant was aspirated, and the pellet resuspended in 5 ml of BSA 2%. The cell suspension was filtered in 40 µm nylon filter and the cells counted in Neubauer chamber. The suspension was centrifuged again and resuspended in NB/B-27 medium and seeded in 50 µg/ml poly-D-lysine (Sigma-Aldrich) and 5 µg/ml laminin (Sigma-Aldrich) coated plates at a density of $2 × 10^5$ cells/well in 24-well plates. The next day, NB medium was changed adding 2 µM of anti-mitotic AraC, and 100 ng/ml of NGF and reducing the concentration of B-27 to 0.2%. Neurons were kept at 37°C in a humidified incubator in a 5% $CO_2$ atmosphere for 11 d for posterior fixation with PFA 2% for 15 min at room temperature. To day 8–10, 10 nM of compound E (Millipore) or DMSO were added to the culture. Cells were permeabilized with 0.1% (vol/vol) Triton X-100/PBS pH 7.4 for 4 min at room temperature and posterior soft denaturalization of 5 min with 0.5% SDS. Coverslips were blocked with 2% BSA for 1 h followed by incubation overnight at 4°C in a humidified chamber with primary goat antibody anti-Choline Acetyltransferase (AB144P; Millipore) 1:200 and rabbit anti-Cleaved Caspase-3 (9661S; Cell Signaling) 1:1,000. Unbound antibody was removed by three washes of PB 0.1M and bound antibody was detected by incubation with Cy3 donkey anti rabbit (Jackson) 1:500, for 1 h or with biotin rabbit anti-goat (Jackson) 1:200 at room temperature for 1 h and posterior cy2 streptavidin (Jackson). Nuclei were stained with DAPI 1:1,000 in PB for 5 min and samples were mounted on glass slides and cover slipped with Mowiol and 50 µl/ml DABCO.

## Supplementary Information

## Acknowledgements

This study was supported by the Spanish Minister of Economy and Competitiveness grant SAF2017-84096-R and by the Generalitat Valenciana 2018-55 to M Vilar. I García-Carpio was supported by an Formación de Personal Investigador (FPI) pre-doctoral fellowship (BFU2013/42746-P) and a mobility grant (EEBB-I-15-10278) from the Spanish Minister of Economy and Competitiveness. This work was funded by the Stichting Alzheimer Onderzoek (S16013) and the Fonds voor Wetenschappelijk Onderzoek or Flanders Research Foundation (FWO) research project (G0B2519N) to L Chávez-Gutiérrez.

## Author Contributions

ML Franco: conceptualization, investigation, and methodology.
I García-Carpio: conceptualization, investigation, methodology, and writing—original draft, review, and editing.
R Comaposada-Baró: investigation and methodology.
JJ Escribano-Saiz: investigation.
L Chávez-Gutiérrez: funding acquisition, investigation, methodology, and writing—original draft, review, and editing.
M Vilar: conceptualization, data curation, supervision, funding acquisition, investigation, methodology, and writing—original draft, review, and editing.

## Conflict of Interest Statement

The authors declare that they have no conflict of interest.

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
