## [Reviewer comments · Life Science Alliance]

Life Science Alliance

TrkA mediated endocytosis of p75-CTF prevents cholinergic neurons death upon γ -secretase inhibition

Maria Franco, Irmina García-Carpio, Raquel Comaposada-Baró, Juan Escribano-Saiz, Lucía Chávez-Gutiérrez, and Marçal Vilar

DOI: <https://doi.org/10.26508/lsa.202000844>

Corresponding author(s): Marçal Vilar, Institute of Biomedicine of Valencia CSIC

Review Timeline:

Submission Date:	2020-07-08
Editorial Decision:	2020-08-20
Revision Received:	2020-11-19
Editorial Decision:	2020-12-16
Revision Received:	2020-12-23
Editorial Decision:	2021-01-06
Revision Received:	2021-01-11
Accepted:	2021-01-11

Scientific Editor: Shachi Bhatt

Transaction Report:

August 20, 2020

Re: Life Science Alliance manuscript #LSA-2020-00844-T

Dr. Marçal Vilar
Institute of Biomedicine of Valencia CSIC
Molecular Basis of Neurodegeneration
C/ Jaume Roig 11
València, Valencia 46010
Spain

Dear Dr. Vilar,

Thank you for submitting your manuscript entitled "Oligomerization of p75CTF prevents its clearance by γ -secretase and drives cholinergic neurons death" to Life Science Alliance (LSA). The manuscript has been reviewed by the editors and outside referees (reviewer comments below). As you will see, the reviewers were quite enthusiastic about the study and its findings, but have raised some concerns that should be addressed prior to further consideration of the manuscript at LSA. Therefore, although we are unable to publish the current version of the manuscript, we would encourage you to submit a revised version that addresses all of the referees' concern, including the role of endocytosis, whether the oligomers/dimers are found in the plasma membrane or endosome, where the activation of TRAF6-JNK pathway happens in the neurons, and all the other clarifications, discussion points, quantifications and better image requests made by the referees.

The revised manuscript maybe re-reviewed, most likely by the original referees. When submitting the revision, please include a letter addressing the reviewers' comments point by point. The typical timeframe for revisions is three months. Please note that papers are generally considered through only one revision cycle, so strong support from the referees on the revised version is needed for acceptance. We would be happy to discuss the individual revision points further with you should this be helpful.

Thank you for this interesting contribution to Life Science Alliance. We are looking forward to receiving your revised manuscript.

Sincerely,
Shachi

Shachi Bhatt
Executive Editor
Life Science Alliance
www.life-science-alliance.org

B. MANUSCRIPT ORGANIZATION AND FORMATTING:

Reviewer #1 (Comments to the Authors (Required)):

In this work the authors aim to demonstrate that inhibition of gamma secretase promotes cell death by inducing oligomerisation of p75 receptors at the plasma membrane, which in turn engages TRAF6, JNK and p38 signalling pathway to trigger apoptosis. TrkA activity is shown to regulate formation of the dimers/oligomers, and then is demonstrated protective against p75-CTF-induced cell death. The draft is well assembled and the experiments are clear and well designed. In particular, the expression of p75-CTF allows to examine specifically the processing step that depends on gamma-secretase, although confirmation of some of the data by using the full length p75 would strengthen the narrative, and the C257A mutation is a valuable tool to manipulate

receptor dimerisation. The use of statistical methods is correct and well applied to the type of data and number of variables.

They present compelling evidence linking inhibition of gamma-secretase and p75-dependent cell death of different cell types, and also showing that oligomeric p75-CTF is less efficiently internalised and somehow protected from the gamma-secretase processing. However there are few critical nodes of the model that have not been clearly supported by the data and discussion of the literature.

Major points

1. Whether TrkA effect on oligomerisation of p75-CTF and cell death depends on endocytosis. It would be perfectly possible that interaction of activated TrkA with p75 leads to higher internalisation, escaping from the locus where cell death signal transduction starts. An experiment like the one in Fig2D-E in the presence of TrkA would help to solve this question.

Whether the dimers/oligomers form exclusively in the plasma membrane, or they can be found in endosomes, isn't clear either. Please discuss your experiments of Fig4 in terms of the role of lipid rafts and cholesterol in endocytosis of p75 as well. Experiments blocking endocytosis would be valuable to tackle this node.

2. Another reason why I would strongly advise the authors to consider emphasising location (plasma membrane versus endosomes) when discussing the mechanism is the fact, mentioned several times in the manuscript, that gamma-secretase processing occurs importantly within endosomal membranes. This spatial dimension is not well integrated into the model and discussion, since everything appears to happen in the surface. What would be the consequences of p75 being sorted to different post-endocytic routes in the presence and absence of TrkA? Would it explain why C257A phenocopies TrkA in terms of rescuing from cell death and blocking oligomerisation?

3. Authors suggest that oligomerisation occurs in membrane domains rich in cholesterol, possibly lipid rafts. Where does the activation of TRAF6-JNK pathway happen in the neuron? There is evidence showing that in sympathetic neurons JNK-pathway is crucial to sort p75 into pro-apoptotic signalling endosomes that propagates from axon terminals to soma. If oligomerisation happens at the plasma membrane, how is the p75-CTF pro-apoptotic signalling propagated? Would the authors say the mechanism operating for long-distance death signals is somehow different?

4. It would be good to discuss other potential mechanisms how TrkA signalling can alter p75 signalling output and dynamics: for example, TrkA-PI3K signalling can turn PIP2 into PIP3, blocking p75-dependent apoptosis and eventually regulating processing as well -as it is regulated by phosphoinositides, as correctly stated in the manuscript.

5. The authors mentioned as a limitation the fact that they worked with a p75-CTF overexpression model, then, excluding from the analysis the potential effect of different ligands. Nevertheless, as for the model this needs to be included in the discussion. What predictions would you do depending on the ligand the neurons have access to? How would ligand availability regulate the role attributed to TrkA in the regulation of gamma-secretase processing of p75? When you suggest that in the clinical trials for gamma-secretase inhibitors p75-dependent pro-apoptotic signalling could have been triggered, are you assuming limited TrkA activity or limited NGF availability?

6. The title suggests that this mechanism is what would lead the cholinergic neurons death after gamma-secretase inhibition. But in the only experiment with cholinergic neurons, inhibition of gamma-secretase doesn't induce cleavage of caspase-3 unless you specifically block TrkA. As discussed before, there is many crosstalk points where TrkA and p75 can counteract each other, including differential sorting, sequestration of the receptor, production of PIP3 (PTEN versus PI3K). At least effects of CE and K252A on oligomerisation need to be shown to support this claim.

Minor Points:

- Please include references for dissection of basal forebrain or detail in the methods.
- In figure 1, legend of section E appears before section D.
- In figure 2E, is not clear what is the comparison which p-values are shown. Please show all the information of the 2-way ANOVA: contribution of p75-CTF WT vs C257A mutant, contribution of time, and interaction between the variables. If multiple comparisons are shown, please compare WT vs mutant. Include exact p-values where it is missing.
- Figure 3F is not included in the legend.
- As for figure 4G-H is understandable why the legend is written in that order, so please compose the figure in a way it corresponds with the order of the legend.
- In page 17, line 7: p75 mutation is misspelled C256A.

Reviewer #2 (Comments to the Authors (Required)):

In this study by Franco et al., the authors provide significant mechanistic insight into how dimerization/oligomerization of the γ -secretase substrate p75 neurotrophin receptor (p75NTR) CTF may induce cell death. The authors determined that γ -secretase inhibition leads to the accumulation of the 75-CTF and p75-CTF dimers and that a p75 dimerization mutant (p75-C257A) induces less cell death. They further determine that oligomerization requires a cholesterol recognition motif (CRAC) and that expression of TrkA reduces p75-CTF dimerization and cell death. Interestingly, TrkA levels are reduced in Alzheimer's disease patients. The authors then delved deeper in the molecular mechanism and determined that expression of p75-CTF and TrkA induce an increase in phosphorylated p38 and JNK levels, although these results were not quantified and were only apparent following inhibition of the γ -secretase. The authors show that TNFR-associated factor 6 (TRAF6), which binds to p75 to regulate JNK-dependent cell death, specifically co-immunoprecipitates with p75-CTF dimers and that TrkA reduces this interaction. Finally, in basal forebrain cholinergic neurons (BFCNs), the authors show that treatment with a GSI and a TrkA inhibitor induces more cell death in wild-type mice relative to BFCNs from p75-KO mice. Collectively, Franco and colleagues provide significant evidence for the p75-CTF pathological cascade involved in neurodegeneration.

Specific points

1. In Fig. 1B, the relevance of the two Western blots panels has not been explained in the figure legend or in the text. For thoroughness of the study, it would be informative to include the dimer formation data under the tested experimental conditions, similar to the figures shown in Fig. 1D and E.
2. In Fig. 1D, it is unclear why DRG neurons were used for this experiment. Please provide an explanation in the text. The results for the HeLa should include a vehicle treated control and should be on the same Western blot as the DRG neurons. The ICD data should also be included in the Western blot,

similar to the blot in Fig. 1E.

3. In Fig. 1E, the p75-CTF accumulation in the presence of the γ -secretase inhibitor (CE) is not apparent in comparison to the data presented Fig. 1D in the same HeLa cell line. Why is this? Please quantify the increase in p75-CTFs and p75-CTF dimer formation in the presence of CE to demonstrate a significant accumulation of the p75-CTFs and dimers.
4. In Fig. 1F, the results would be strengthened by showing equivalent expression of the different constructs (FL, CTF, ICD) by Western blot analysis. Why were cortical neurons used here and DRG neurons in Fig. 1D?
5. In Fig. 1G, please show representative images for all conditions.
6. In Fig. 2A, it would be informative to show the p75-CTF dimer bands under these conditions for comparison with Fig. 2C.
7. In Fig. 2B, are the results statistically significant?
8. In Fig. 2C, for accurate comparison, the starting protein concentration of the monomer CTF and CTF-257A at time 0 should be comparable.
9. In Fig. 2E, it is unclear how many times the experiment was performed, the number of replicates, and how the quantification was performed. Please add this information to the figure legend.
10. In Fig. 4, why are the levels of p75-CTF dimers so low in the p75-CTF transfected HeLa cells in Fig. 4E and 4F? Levels are much more appreciable in Fig. 1D and 1E without treatment.
11. In Fig. 4F, in the absence of BS3, is the p75 CTF dimer/monomer ratio in p75CTF relative to p75CTF+TrkA significant? Same comment for Fig. 4G.
12. In Fig. 4I, is the increase in %caspase-3+ cells in p75-CTF alone relative to control significant? Is the difference between p75-CTF alone and reduction in p-75-CTF+TrkA significant? To strengthen and support the results, please show the immunofluorescence staining and show Western blot data of equivalent expression of each construct.
13. In Fig. 5, it is not indicated how many times the experiments were performed or the number of replicates per experiment. What happens to the p75-CTF dimers in this experiment? Do they correspond with an increase in p-p38?
14. In Fig. 5C, please show the full-length p75 and the p75-CTF dimers.
15. What is the difference between the blots in 5g and 5I? One blot with p75-FL, CTF, and dimers would be sufficient.
16. In Fig. 6B, please provide quantitative data for the changes in p-p38 and p-JNK levels. Why are the increased band intensities only visible following CE treatment?
17. In Fig. 6D, please show a Western blot demonstrating the reduction in p75CTF dimer interaction with TRAF6 following co-expression of TrkA+NGF treatment.

18. In Fig. 7, please show images of the treatment conditions in Fig. 7B.

Minor points

1. Fig. 1A is not referred to in the text.
2. Please add scale to Figure 1G Y-axis.
3. Fig. 5C should be Fig. 5A. Please place the figures in the order in which they are discussed in the text.

Reviewer #1 (Comments to the Authors (Required)):

In this work the authors aim to demonstrate that inhibition of gamma secretase promotes cell death by inducing oligomerisation of p75 receptors at the plasma membrane, which in turn engages TRAF6, JNK and p38 signalling pathway to trigger apoptosis. TrkA activity is shown to regulate formation of the dimers/oligomers, and then is demonstrated protective against p75-CTF-induced cell death. The draft is well assembled and the experiments are clear and well designed. In particular, the expression of p75-CTF allows to examine specifically the processing step that depends on gamma-secretase, although confirmation of some of the data by using the full length p75 would strengthen the narrative, and the C257A mutation is a valuable tool to manipulate receptor dimerisation. The use of statistical methods is correct and well applied to the type of data and number of variables.

They present compelling evidence linking inhibition of gamma-secretase and p75-dependent cell death of different cell types, and also showing that oligomeric p75-CTF is less efficiently internalised and somehow protected from the gamma-secretase processing. However there are few critical nodes of the model that have not been clearly supported by the data and discussion of the literature.

Major points

1. Whether TrkA effect on oligomerisation of p75-CTF and cell death depends on endocytosis. It would be perfectly possible that interaction of activated TrkA with p75 leads to higher internalisation, escaping from the locus where cell death signal transduction starts. An experiment like the one in Fig2D-E in the presence of TrkA would help to solve this question. Whether the dimers/oligomers form exclusively in the plasma membrane, or they can be found in endosomes, isn't clear either. Please discuss your experiments of Fig4 in terms of the role of lipid rafts and cholesterol in endocytosis of p75 as well. Experiments blocking endocytosis would be valuable to tackle this node.

We thank the reviewer for this valuable input. We have performed a set of different experiments in different cell lines like HEK293 and in PC12 and PC12nnr5 cells, to demonstrate that TrkA effectively promotes p75CTF internalization (new Figure 2).

We also did the experiment using flow cytometry (Figure S3). We found that,

1.- TrkA induces the internalization of p75CTF in a NGF-dependent manner (new Figures 2D to 2H)

2.- inhibition of TrkA activity with K252a or the inhibition of macropinocytosis (amiloride treatment) inhibits the internalization of p75-CTF (Figure 2F).

3.- internalization of p75CTF correlates with less crosslinked oligomers and less cell death (Figures 4 and 5), suggesting that TrkA-mediated internalization of p75-CTF is one of the most important mechanism to avoid p75CTF toxicity.

Interestingly we also found that the mutant p75CTF-C257A is differently internalized in the presence of TrkA (Figure S2), suggesting that the oligomerization state or the different subcellular distribution of p75-CTF-wt vs p75CTF-C257A play a role in its internalization promoted by TrkA/NGF. Recently a pre-print in [biorxiv.org](https://doi.org/10.1101/2020.01.10.901926) from the Ibañez laboratory (bioRxiv 2020.01.10.901926; doi: <https://doi.org/10.1101/2020.01.10.901926>) showed that p75C257A is less internalized than p75-wt in hippocampal neurons (that express TrkB), supporting our results showing the differential internalization between p75CTF-wt and p75CTF-C257A in the presence of TrkA.

Regarding the question of the location of p75CTF oligomers we can say that as we used BS3 crosslinking agent and this reagent cannot cross the plasma membrane we can be sure that oligomers form at the plasma membrane. Nevertheless we agree with the reviewer that the formation of p75CTF oligomers could also be formed or remained in the internalized endosomes as well, so we further discuss this possibility in the text in the Discussion section.

Regarding the role of cholesterol in p75 endocytosis we add the following line in the discussion text (Page 19, lines 424-432) "Cholesterol rich domains also play a role in the receptor internalization. It is known that in neurons p75 could be internalized through clathrin-dependent and clathrin-independent pathways depending of the presence of ligand neurotrophin, each one leading to different sorting pathways; like receptor recycling or axonal transport (Deinhardt *et al*, 2007; Bronfman *et al*, 2003). In this sense the finding that oligomerization of p75CTF is modulated by cholesterol could be related to the targeting of these oligomers to specific plasma membrane locations where internalization and the sequential sorting to different internalized endosomes would take place."

2. Another reason why I would strongly advise the authors to consider emphasising location (plasma membrane versus endosomes) when discussing the mechanism is the fact, mentioned several times in the manuscript, that gamma-secretase processing occurs importantly within endosomal membranes. This spatial dimension is not well integrated into the model and discussion, since everything appears to happen in the surface. What would be the consequences of p75 being sorted to different post-endocytic routes in the presence and absence of TrkA? Would it explain why C257A phenocopies TrkA in terms of rescuing from cell death and blocking oligomerisation?

We thank the reviewer for this comment. We have further discussed the spatial dimension in the new version of the manuscript (see Discussion lines 424-478) and we have refined our model in Figure 8 to reflect this possibility.

How p75CTF is internalized by itself is a question that needs to be studied in more detail in the near future, but our data indicates that p75CTF and p75CTF-C57A should be differentially sorted once internalized because they induced different phenotypes. Our results suggest that the formation of high-molecular weight oligomers and the increase of PIP2 levels are behind this different sorting. As TrkA is able to modulate the formation of such oligomers and the levels of PIP2, it could modulate the final fate of p75CTF-containing endosomes. What is the fate and function of internalized endosomes containing p75CTF in the absence or presence of TrkA is a very interesting topic and not studied here, although we speculate that TrkA directs p75CTF to signaling endosomes (by activating macropinocytosis) and p75CTF is sorted to a different kind of endosomes (apoptotic endosome?). In any case we discuss the various possibilities in the Discussion section (Discussion lines 479-511).

Regarding that TrkA phenocopies p75CTF-C257A, we think they used different mechanisms, as p75CTFC257A is quite unstable in the cell and is rapidly degraded (probably by being sorted to lysosomes). By contrast TrkA activity promotes p75CTF internalization but does not induce an increase of p75CTF degradation. The function of these endosomes is still in debate but they could be forming part of a special type of signaling endosomes with a specific function.

3. Authors suggest that oligomerisation occurs in membrane domains rich in cholesterol, possibly lipid rafts. Where does the activation of TRAF6-JNK pathway happen in the neuron? There is evidence showing that in sympathetic neurons JNK-pathway is crucial to sort p75 into pro-apoptotic signalling endosomes that propagates from axon terminals to soma. If oligomerisation happens at the plasma membrane, how is the p75-CTF pro-

apoptotic signalling propagated? Would the authors say the mechanism operating for long-distance death signals is somehow different?

We are aware of those studies and following the previous comments we now discuss the possibility to have p75CTF oligomers in the internalized endosomes (see the new Discussion lines 424-511). Although comparing our system with the sympathetic neurons is difficult and risky, we must note that our data should be put into context of a special situation described in old patients of AD treated with GSIs (γ -secretase inhibition and TrkA activity downregulated) and we are not sure that the mechanism of cell death in BFCNs we described here is the same as the one found in physiological conditions during development, as is the case for sympathetic neurons cell death stimulated with BDNF. Nevertheless it is known that TRAF6 mediates p75 ubiquitination and receptor internalization. It could be possible that oligomerization of p75CTF might nucleate TRAF6 binding and the activation of the JNK pathway similarly to sympathetic neurons stimulated with BDNF (Escudero *et al.* 2018) may induce their internalization. Alternatively it could be possible that internalized p75CTF oligomers could be still recruiting TRAF6 and activate the JNK pathway. The binding of TRAF6 to internalized endosomes is an option already described in other TNFRs. Recently a co-localization of TRAF6 with rab7-positive endosomes in CD40 receptor signalling has been described (Yan *et al.*, 2020). As p75 has been found to co-localize to Rab7-positive endosomes in retrograde axonal transport (Deinhardt *et al.*, 2007) it could be possible that p75/TRAF6 complex form part of the pro-apoptotic endosomes, a hypothesis that should be tested in the future. This has been added in the Discussion section of the new revised manuscript.

4. It would be good to discuss other potential mechanisms how TrkA signalling can alter p75 signalling output and dynamics: for example, TrkA-PI3K signalling can turn PIP2 into PIP3, blocking p75-dependent apoptosis and eventually regulating processing as well -as it is regulated by phosphoinositides, as correctly stated in the manuscript.

We thank the referee for this comment. It has been shown that overexpression of p75CTF activate PIP-5-K and increase the levels of PIP2. As PIP2 play an important role in endocytosis, during the revision of this manuscript we carried out a set of experiments to determine if the modulation of the PIP2 levels may be behind the promotion of p75CTF endocytosis by TrkA/NGF. We co-express the PIP2 phosphatase synaptojanin that downregulates the total levels of PIP2, and found that synaptojanin increases p75CTF internalization and reduces p75CTF oligomerization (new Figure 5H and S4) phenocopying the role of TrkA/NGF. As TrkA is able to downregulate the levels of PIP2 by the activation of PI3K or the PLC γ , this experiment suggested that TrkA regulation of PIP2 levels may be behind its promotion of p75CTF internalization and reduction of p75CTF at the plasma membrane. This is discussed in the Discussion section lines 433-454 and in the Figure 8.

5. The authors mentioned as a limitation the fact that they worked with a p75-CTF overexpression model, then, excluding from the analysis the potential effect of different ligands. Nevertheless, as for the model this needs to be included in the discussion. What predictions would you do depending on the ligand the neurons have access to? How would ligand availability regulate the role attributed to TrkA in the regulation of gamma-secretase processing of p75? When you suggest that in the clinical trials for gamma-secretase inhibitors p75-dependent pro-apoptotic signalling could have been triggered, are you assuming limited TrkA activity or limited NGF availability?

We totally agree with the reviewer that the role of different ligands on p75 shedding is very important because they will determine the outcome of the cell and following the recommendation of the reviewer we have included the role of pro-NGF or NGF in our hypothesis (see Discussion lines 525-540).

Regarding our comment on the clinical trials of GSIs in the older AD patients there is a reduction in the expression of TrkA in the BFCNs and less activation and an increase in the levels of Pro-NGF that is not able to activate TrkA as efficiently as NGF. What we suggested here is that the lower levels of TrkA expression and activation together with and diminished function of the γ -secretase due to familiar mutations in the g-secretase complex or with the use of GSIs, would lead to increased p75CTF oligomerization (mainly generated by the constitutive or increased shedding of p75) and cell death of BFCNs. This is added in the Discussion section (lanes 528-540).

6. The title suggests that this mechanism is what would lead the cholinergic neurons death after gamma-secretase inhibition. But in the only experiment with cholinergic neurons, inhibition of gamma-secretase doesn't induce cleavage of caspase-3 unless you specifically block TrkA. As discussed before, there is many crosstalk points where TrkA and p75 can counteract each other, including differential sorting, sequestration of the receptor, production of PIP3 (PTEN versus PI3K). At least effects of CE and K252A on oligomerisation need to be shown to support this claim.

We agree with the referee that to have a toxic effect on BFCNs both the g-secretase and the TrkA activity need to be inhibited. This has been now better integrated in the text and in the Discussion section. After this round of revision and based on the new data we provide in this new manuscript we are confident that the role of TrkA affects different points of regulation, like p75CTF oligomerization, modulation of PIP2 levels and its internalization including probably a different sorting. Based on this we decided to modify the title of the manuscript to include this complexity of roles attributed to TrkA. The new title "*TrkA mediated endocytosis of p75-CTF prevents cholinergic neurons death upon γ -secretase inhibition*".

On the other hand we hardy tried to do some biochemical characterization to identify the oligomers of p75 in BFCNs in culture. Unfortunately those experiments were challenging due to the low yield of actual cholinergic neurons we get from the basal forebrain (only 10-15% live neurons from the basal forebrain are ChAT+/p75+ neurons as reported previously (Schnitzler *et al*, 2008)). Thus as a approximation we did the experiment in PC12 cells that express endogenous levels of p75 and TrkA and found that incubation with CE lead to the oligomerization of p75 only in PC12nnr5 cells that do not express TrkA cells but not in PC12 cells (new Figure 4C).

Minor Points:

- Please include references for dissection of basal forebrain or detail in the methods. We included the main reference (Schnitzler *et al*, 2008) in the method section.
- In figure 1, legend of section E appears before section D. We changed the order in the writing.
- In figure 2E, is not clear what is the comparison which p-values are shown. Please show all the information of the 2-way ANOVA: contribution of p75-CTF WT vs C257A mutant, contribution of time, and interaction between the variables. If multiple comparisons are shown, please compare WT vs mutant. Include exact p-values where it is missing. We added the new Figure 2E with a two-way ANOVA analysis in the text. Now in the main text we add "(Two-way ANOVA analysis; time factor $F(5,10)=335.3$ $P<0.0001$; mutant factor $F(3,6)=46.91$, $P=0.0001$; both factors $F(11,74)=15.30$ $P<0.0001$)"
- Figure 3F is not included in the legend.

The legend of Figure 3 is complete now.

- As for figure 4G-H is understandable why the legend is written in that order, so please compose the figure in a way it corresponds with the order of the legend.

We thank the reviewer for this point. We corrected the legend accordingly. Please take into account that the old Figure 4 is now Figure 5 and vice versa.

- In page 17, line 7: p75 mutation is misspelled C256A.

We corrected that misspelling. Thank you.

Reviewer #2 (Comments to the Authors (Required)):

In this study by Franco et al., the authors provide significant mechanistic insight into how dimerization/oligomerization of the γ -secretase substrate p75 neurotrophin receptor (p75NTR) CTF may induce cell death. The authors determined that γ -secretase inhibition leads to the accumulation of the 75-CTF and p75-CTF dimers and that a p75 dimerization mutant (p75-C257A) induces less cell death. They further determine that oligomerization requires a cholesterol recognition motif (CRAC) and that expression of TrkA reduces p75-CTF dimerization and cell death. Interestingly, TrkA levels are reduced in Alzheimer's disease patients. The authors then delved deeper in the molecular mechanism and determined that expression of p75-CTF and TrkA induce an increase in phosphorylated p38 and JNK levels, although these results were not quantified and were only apparent following inhibition of the γ -secretase. The authors show that TNFR-associated factor 6 (TRAF6), which binds to p75 to regulate JNK-dependent cell death, specifically co-immunoprecipitates with p75-CTF dimers and that TrkA reduces this interaction. Finally, in basal forebrain cholinergic neurons (BFCNs), the authors show that treatment with a GSI and a TrkA inhibitor induces more cell death in wild-type mice relative to BFCNs from p75-KO mice. Collectively, Franco and colleagues provide significant evidence for the p75-CTF pathological cascade involved in neurodegeneration.

Specific points

1. In Fig. 1B, the relevance of the two Western blots panels has not been explained in the figure legend or in the text. For thoroughness of the study, it would be informative to include the dimer formation data under the tested experimental conditions, similar to the figures shown in Fig. 1D and E.

Thank you for this valuable suggestion. We have simplified Figure 1B and we now show only a single Western blot panel. The quantification of all the conditions from this panel is shown in Figure 1C. The panel that presented the results from the NH₄Cl treatment has been moved to Figure S1.

Regarding dimer formation under the Figure 1B experimental conditions, please note that we cannot detect the dimer because it is a reducing SDS-PAGE. This information has been added in the Figure legend. Please note as well that the aim of Figure 1B and 1C is to explore the contribution of different degradation pathways to p75 proteolysis.

2. In Fig. 1D, it is unclear why DRG neurons were used for this experiment. Please provide an explanation in the text. The results for the HeLa should include a vehicle treated control and should be on the same Western blot as the DRG neurons. The ICD data should also be included in the Western blot, similar to the blot in Fig. 1E.

DRG neurons were chosen because they express p75 and TrkA at endogenous levels. This is now mentioned in the text (page 7, lines 132-133).

In the previous version of the manuscript we included a HeLa western blot panel next to the DRG panel to illustrate the mobility of the p75CTF monomer and dimer. However, we agree that this is not adequate. Since the position of the monomer and dimer in HeLa

extracts is widely shown throughout the manuscript, we have decided to remove the panel. Also, we have now labeled the position corresponding to p75^{full-length} on the DRG blot, in addition to the p75CTF monomer and p75CTF dimer, to clarify the identity of the bands. Please note that the ICD cannot be detected in the Western blot since the DRGs were not treated with epoxomycin. We hope the reviewer agrees with our changes.

3. In Fig. 1E, the p75-CTF accumulation in the presence of the γ -secretase inhibitor (CE) is not apparent in comparison to the data presented Fig. 1D in the same HeLa cell line. Why is this? Please quantify the increase in p75-CTFs and p75-CTF dimer formation in the presence of CE to demonstrate a significant accumulation of the p75-CTFs and dimers.

Please notice that this blot is from lysates from purified membrane fractions and not from total lysates. In any case we have now quantified the blot shown in the new Figure 1D and the bar plot is the new Figure 1E showing the increase of the p75CTF dimer/monomer ratio.

4. In Fig. 1F, the results would be strengthened by showing equivalent expression of the different constructs (FL, CTF, ICD) by Western blot analysis. Why were cortical neurons used here and DRG neurons in Fig. 1D?

We agree with the reviewer. We now present the immunoprecipitation data of the cortical neurons lysates with a p75 specific antibody (against the ICD of p75). The western blot (new Figure 1I) shows the equivalent expression of the different constructs in the absence of CE. Please note that the light chain of the antibody migrates close to the p75CTF protein band.

Cortical neurons were chosen in this experiment instead of DRG neurons because they do not express significant levels of p75. These neurons are a suitable model because we can study the role of transfected p75CTF and p75CTF-C257A constructs in the absence of any contribution from the endogenous p75. This is now mentioned in the main text.

5. In Fig. 1G, please show representative images for all conditions.

We have added a representative immunofluorescence image from all conditions in the new Figure 1G.

6. In Fig. 2A, it would be informative to show the p75-CTF dimer bands under these conditions for comparison with Fig. 2C.

In reducing conditions the dimers are not visible. In the non-reducing conditions (Figure 2C) the position of the p75CTF dimers are indicated.

7. In Fig. 2B, are the results statistically significant?

We performed a 2-way ANOVA analysis and the results are shown in the Figure 2B and in the main text (page 8, lines 154-156).

8. In Fig. 2C, for accurate comparison, the starting protein concentration of the monomer CTF and CTF-257A at time 0 should be comparable.

We understand the concern of the reviewer but the two mutants showed different protein degradation so never they showed the same levels of protein concentration at time 0. In any case we always take into account this difference and the quantification is normalized by the protein levels of each mutant at the time 0.

9. In Fig. 2E, it is unclear how many times the experiment was performed, the number of replicates, and how the quantification was performed. Please add this information to the

figure legend.

We performed a 2-way ANOVA analysis and the results are shown in the Figure 2E and in the main text (page 8, lines 167-169).

10. In Fig. 4, why are level of p75-CTF dimers so low in the p75-CTF transfected HeLa cells in Fig. 4E and 4F? Levels are much more appreciable in Fig. 1D and 1E without treatment.

This is a consequence of the western blot exposition intensity to show differences between treated and untreated samples.

11. In Fig. 4F, in the absence of BS3, is the p75 CTF dimer/monomer ratio in p75CTF relative p75CTF+TrkA significant? Same comment for Fig. 4G.

We performed a 2-way ANOVA analysis and the results are shown in the Figure 5D and in the main text (page 13, lines 276-279). Please note that in the new version of the manuscript the old Figure 4 is the new Figure 5.

12. In Fig. 4I, is the increase in %caspase-3+ cells in p75-CTF alone relative to control significant? Is the difference between p75-CTF alone and reduction in p-75-CTF+TrkA significant? To strengthen and support the results, please show the immunofluorescence staining and show Western blot data of equivalent expression of each construct.

We performed a 2-way ANOVA analysis and the results are shown in the Figure S5 and in the main text (page 14, lines 317-319). Please note that in the new version of the manuscript the old Figure 4 is the new Figure 5.

Representative blots are shown in the Figure 5I and a representative image of the immunofluorescence is shown in Figure S5.

13. In Fig. 5, it is not indicated how many times the experiments were performed or the number of replicates per experiment. What happens to the p75-CTF dimers in this experiment? Do they correspond with an increase in p-p38?

14. In Fig. 5C, please show the full-length p75 and the p75-CTF dimers.

We combined the two questions in one as they refer to the same topic. Please note that in the new version of the manuscript the old Figure 5 is the new Figure 4. The number of experiments is now indicated in the Figure 4 legend. Now it is shown a western blot showing the formation of p75 oligomers crosslinked by BS3 in PC12nnr5 cells but not in PC12 cells (new Figure 4C). The formation of oligomers correlate with an increase in P-p38 as shown in the new Figures 4D-4E.

15. What is the difference between the blots in 5g and 5I? One blot with p75-FL, CTF, and dimers would be sufficient.

We understand that the old 5G and 5I blots were redundant. Now we show only one blot in Figure 4C. Please note that in the new version of the manuscript the old Figure 5 is the new Figure 4.

16. In Fig. 6B, please provide quantitative data for the changes in p-p38 and p-JNK levels. Why are the increased band intensities only visible following CE treatment?

We now show the quantification of the levels of p-p38 and p-JNK in the new Figure 6C. We think that CE exacerbates the formation of oligomers and these trigger p38 activation. Alternatively the inhibition of the g-secretase complex could induce an activation of p38, but that only takes place upon overexpression of p75 in the absence of TrkA.

17. In Fig. 6D, please show a Western blot demonstrating the reduction in p75CTF dimer interaction with TRAF6 following co-expression of TrkA+NGF treatment.

We performed the co-immunoprecipitation of TRAF6 in the presence of p75CTF and TrkAplus NGF (new Figure 6F). Experiments showed that TrkA displaces TRAF6 from the binding to p75CTF dimers. We do not detect an effect of NGF in this interaction probably because overexpression of TrkA induces its autoactivation independent of ligand.

18. In Fig. 7, please show images of the treatment conditions in Fig. 7B.

We now incorporated representative immunofluorescence images of all the conditions analyzed in the Figure 7.

Minor points

1. Fig. 1A is not referred to in the text.

The Figure 1A is referenced in the Introduction (page 4, line 62) lines as a scheme to present that p75 suffers different processing events.

2. Please add scale to Figure 1G Y-axis.

The scale has been added in the Figure.

3. Fig. 5C should be Fig. 5A. Please place the figures in the order in which they are discussed in the text.

The order has been corrected.

Bibliography

- Bronfman FC, Tcherpakov M, Jovin TM & Fainzilber M (2003) Ligand-induced internalization of the p75 neurotrophin receptor: a slow route to the signaling endosome. *J. Neurosci.* **23**: 3209–3220
- Deinhardt K, Reversi A, Berninghausen O, Hopkins CR & Schiavo G (2007) Neurotrophins Redirect p75NTR from a clathrin-independent to a clathrin-dependent endocytic pathway coupled to axonal transport. *Traffic* **8**: 1736–1749
- Schnitzler AC, Lopez-Coviella I & Blusztajn JK (2008) Purification and culture of nerve growth factor receptor (p75)-expressing basal forebrain cholinergic neurons. *Nat. Protoc.* **3**: 34–40
- Yan H, Fernandez M, Wang J, Wu S, Wang R, Lou Z, Moroney JB, Rivera CE, Taylor JR, Gan H, Zan H, Kolvaskyy D, Liu D, Casali P & Xu Z (2020) B Cell Endosomal RAB7 Promotes TRAF6 K63 Polyubiquitination and NF- κ B Activation for Antibody Class-Switching. *J. Immunol.* **204**: 1146–1157

December 16, 2020

RE: Life Science Alliance Manuscript #LSA-2020-00844-TR

Dr. Marçal Vilar
Institute of Biomedicine of Valencia CSIC
Molecular Basis of Neurodegeneration
C/ Jaume Roig 11
València, Valencia 46010
Spain

Dear Dr. Vilar,

Thank you for submitting your revised manuscript entitled "TrkA mediated endocytosis of p75-CTF prevents cholinergic neurons death upon γ -secretase inhibition".

As you will see from the reviewers' comments below, the reviewers are quite happy with the revised manuscript, but do think that some minor edits are required before the manuscript is ready for publication. We would be happy to publish your paper in Life Science Alliance pending final revisions as pointed out by the reviewers and further edits necessary to meet our formatting guidelines.

Along with the points listed below, please also attend to the following:

- please consult our Manuscript Preparation Guidelines <https://www.life-science-alliance.org/manuscript-prep> and put your manuscript sections in the correct order
- please use the [10 author names, et al.] format in your references (i.e. limit the author names to the first 10)
- please provide the source data (uncropped, unedited gel images) for Figure 1I, and Figure 6B
- please add the legend for Figure 2H
- please make sure that the insets matched the respective zoomed in panels in Figure 2D (rows 1-3), Figure 2F, Figure 2H (bottom row), and Figure S4A
- since there are no other panels in Figure S5, the image and the legend do not need to have a 'panel A'

A. FINAL FILES:

B. MANUSCRIPT ORGANIZATION AND FORMATTING:

Sincerely,

Shachi Bhatt, Ph.D.
Executive Editor

Reviewer #1 (Comments to the Authors (Required)):

In this work the authors aim to demonstrate that when TrkA signalling is blocked, inhibition of gamma secretase promotes cell death by inducing oligomerisation of p75 receptors at the plasma membrane, which in turn engages TRAF6, JNK and p38 signalling pathway to trigger apoptosis. TrkA activity is shown to regulate formation of the dimers/oligomers through several mechanisms, including endocytosis of p75-CTF, phosphoinositides membrane composition and intracellular signalling, and then is demonstrated protective against p75-CTF-induced cell death. The draft is well assembled and the experiments are clear and well designed. In particular, the expression of p75-CTF allows to examine specifically the processing step that depends on gamma-secretase, although confirmation of some of the data by using the full length p75 would strengthen the narrative, and the C257A mutation is a valuable tool to manipulate receptor dimerisation. The use of statistical methods is correct and well applied to the type of data and number of variables.

In the original manuscript, they presented compelling evidence linking inhibition of gamma-secretase and p75-dependent cell death of different cell types, and also showing that oligomeric p75-CTF is less efficiently internalised and somehow protected from the gamma-secretase processing. After this revision the authors added meaningful experiments and re-oriented the model to integrate the role of TrkA signalling in endocytosis, and trying to account for the location where the processing of p75-CTF and the apoptotic signalling occur, as well as for the propagation of the apoptotic signal onboard of specialised signalling endosomes. I'm very happy with the significant expansion of the experimental data and the new shape that the article took after the revision. I'm glad to see that our comments were carefully taken onboard and led the authors to significantly improve and refine their work from the very title itself.

I have only few comments that in my opinion should be taken into account before this work is ready for publication:

1. This new orientation that includes the crucial role of TrkA-regulated internalisation of p75 is missing from the ending of the introduction where the findings are summarised.
2. Figure 1E has no error bars. Is not clear if the experiment was done only once. Please make sure the number of replicates is indicated for all the experiments and plots.
3. In Figure 1G-H a role of cystein 257 is argued based on the apparent decrease of cleaved caspase 3 levels that CE induces in cells expressing the C257A mutant, but this is not clear from the data, essentially for 2 reasons: i) in p75-CTF-C257A neurons cleaved caspase 3 is lower than p75-CTF-WT neurons in both with and without CE, suggesting that this decrease is independent of the role of CE; ii) the decrease is similar to what you observe when expressing p75-FL-WT and still significantly higher than EV control.
4. In the Experimental Procedures the conditions for maintenance of BFCN is indicated under the subtitle "Cell lines culture" which is incorrect and redundant, giving that you then describe the whole culture. In both sections the amount of NGF supplemented to the media is stated to be 100 ng,

please indicate the concentration instead (I'm assuming it must say 100 ng/mL, which is rather high by the way).

5. In the Discussion section, line 478, it is asserted that your results show that dimeric p75-CTF is not cleaved by γ -secretase "in endogenous conditions". Given that the findings were done in cell lines overexpressing a truncated receptor, it would be better to avoid that expression and replace it with something more like "in a cellular context".

Reviewer #2 (Comments to the Authors (Required)):

In the revised manuscript, Franco et al. have addressed all of the issues raised by this reviewer from the original version. Overall, the authors have made the revised manuscript more compelling. However, I have a few minor queries.

1. In Fig. 1I, the quality of the Western blot showing the p75-CTF is poor and appears to be cut just above the 25 kDa marker. Please provide a better quality/more convincing blot from the 3 experiments. It is also difficult to visualize the dimers in this Western blot.

2. The figure legend states DR were transfected in Fig. 1I. However, the text states that cortical neurons were used for these experiments. Please clarify.

3. In Fig. 1G and 4F, how many fields of view were visualized to generate the data? Approximately how many cells?

Letter to the Reviewers,

First of all we thank the reviewers of this manuscript for their advices and comments on our manuscript. We think the manuscript now is better than the first version.

In this work the authors aim to demonstrate that when TrkA signalling is blocked, inhibition of gamma secretase promotes cell death by inducing oligomerisation of p75 receptors at the plasma membrane, which in turn engages TRAF6, JNK and p38 signalling pathway to trigger apoptosis. TrkA activity is shown to regulate formation of the dimers/oligomers through several mechanisms, including endocytosis of p75-CTF, phosphoinositides membrane composition and intracellular signalling, and then is demonstrated protective against p75-CTF-induced cell death. The draft is well assembled and the experiments are clear and well designed. In particular, the expression of p75-CTF allows to examine specifically the processing step that depends on gamma-secretase, although confirmation of some of the data by using the full length p75 would strengthen the narrative, and the C257A mutation is a valuable tool to manipulate receptor dimerisation. The use of statistical methods is correct and well applied to the type of data and number of variables.

In the original manuscript, they presented compelling evidence linking inhibition of gamma-secretase and p75-dependent cell death of different cell types, and also showing that oligomeric p75-CTF is less efficiently internalised and somehow protected from the gamma-secretase processing. After this revision the authors added meaningful experiments and re-oriented the model to integrate the role of TrkA signalling in endocytosis, and trying to account for the location where the processing of p75-CTF and the apoptotic signalling occur, as well as for the propagation of the apoptotic signal onboard of specialised signalling endosomes. I'm very happy with the significant expansion of the experimental data and the new shape that the article took after the revision. I'm glad to see that our comments were carefully taken onboard and led the authors to significantly improve and refine their work from the very title itself.

I have only few comments that in my opinion should be taken into account before this work is ready for publication:

1. This new orientation that includes the crucial role of TrkA-regulated internalisation of p75 is missing from the ending of the introduction where the findings are summarised.

We added a sentence reflecting this at the end of the introduction section.

2. Figure 1E has no error bars. Is not clear if the experiment was done only once. Please make sure the number of replicates is indicated for all the experiments and plots.

We added the number of replicates in all figure legends.

3. In Figure 1G-H a role of cystein 257 is argued based on the apparent decrease of cleaved caspase 3 levels that CE induces in cells expressing the C257A mutant, but this is not clear from the data, essentially for 2 reasons: i) in p75-CTF-C257A neurons cleaved caspase 3 is lower than p75-CTF-WT neurons in both with and without CE, suggesting that this decrease is independent of the role of CE; ii) the decrease is similar to what you observe when expressing p75-FL-WT and still significantly higher than EV control.

We agree with the referee that the differences between p75-CTF and p75-CTF-C257A are low, but they are significant statistically. In any case we lower the tone a bit, and change the sentence to *"The significant increment in cell death observed in p75-CTF-wt overexpressing neurons relative to the mutant, suggests a partial, but significant, contribution of the cysteine residue to the p75-CTF-mediated toxicity"*.

4. In the Experimental Procedures the conditions for maintenance of BFCN is indicated under the subtitle "Cell lines culture" which is incorrect and redundant, giving that you then describe the whole culture. In both sections the amount of NGF supplemented to the media is stated to be 100 ng, please indicate the concentration instead (I'm assuming it must say 100 ng/mL, which is rather high by the way).

We add the changes requested.

5. In the Discussion section, line 478, it is asserted that your results show that dimeric p75-CTF is not cleaved by γ -secretase "in endogenous conditions". Given that the finding were done in cell lines overexpressing a truncated receptor, it would be better to avoid that expression and replace it with something more like "in a cellular context".

We changed the sentence to *"Our studies show for the first time that γ -secretase is not able to cleave a naturally dimeric p75-CTF substrate"*.

Reviewer #2 (Comments to the Authors (Required)):

In the revised manuscript, Franco et al. have addressed all of the issues raised by this reviewer from the original version. Overall, the authors have made the revised manuscript more compelling. However, I have a few minor queries.

1. In Fig. 1I, the quality of the Western blot showing the p75-CTF is poor and appears to be cut just above the 25 kDa marker. Please provide a better quality/more convincing blot from the 3 experiments. It is also difficult to visualize the dimers in this Western blot.

The blot is not cropped and it is a full gel. We provide a higher resolution uncropped blot. Dimers of p75CTF are not visible as the gel is a normal reducing SDS-PAGE.

2. The figure legend states DR were transfected in Fig. 1I. However, the text states that cortical neurons were used for these experiments. Please clarify.

The experiment was done in cortical neurons. The mistake has been corrected in the legend

3. In Fig. 1G and 4F, how many fields of view were visualized to generate the data?
Approximately how many cells?

In the Figure 1I, cortical neurons were electroporated with a nucleofector. Every GFP+ neurons was counted in every well and condition. The experiment was done per triplicate. In total around 50 neurons were counted per condition.

In the Figure 4F, more than 500 GFP+PC12 cells were counted per condition and experiment.

This information has been added in the figure legend.

January 6, 2021

RE: Life Science Alliance Manuscript #LSA-2020-00844-TRR

Dr. Marçal Vilar
Institute of Biomedicine of Valencia CSIC
Molecular Basis of Neurodegeneration
C/ Jaume Roig 11
València, Valencia 46010
Spain

Dear Dr. Vilar,

Thank you for submitting your revised manuscript entitled "TrkA mediated endocytosis of p75-CTF prevents cholinergic neurons death upon γ -secretase inhibition", and making the formatting edits.

I think there are few things still missing - the updated supplemental figure numbers are not reflected in the manuscript text callouts. I am sending this back to you so you can fix it and re-send me the manuscript that we can then accept.

- callouts for Figures S2 A and B and S3 A and B missing
- can you please change the labels on the actual graphics of the supplemental figures, so when we open Supplementary figure 2 it is labeled as S2 and not S3.. similar for S3 and S4.
- there is a mention of panel A in Legend for Figure S4, although there are no panels in the figure

Sincerely,

Shachi Bhatt, Ph.D.
Executive Editor
Life Science Alliance

<https://www.lsjournal.org/>
Tweet @SciBhatt @LSAJournal

Letter to the Reviewers,

First of all we thank the reviewers of this manuscript for their advices and comments on our manuscript. We think the manuscript now is better than the first version.

In this work the authors aim to demonstrate that when TrkA signalling is blocked, inhibition of gamma secretase promotes cell death by inducing oligomerisation of p75 receptors at the plasma membrane, which in turn engages TRAF6, JNK and p38 signalling pathway to trigger apoptosis. TrkA activity is shown to regulate formation of the dimers/oligomers through several mechanisms, including endocytosis of p75-CTF, phosphoinositides membrane composition and intracellular signalling, and then is demonstrated protective against p75-CTF-induced cell death. The draft is well assembled and the experiments are clear and well designed. In particular, the expression of p75-CTF allows to examine specifically the processing step that depends on gamma-secretase, although confirmation of some of the data by using the full length p75 would strengthen the narrative, and the C257A mutation is a valuable tool to manipulate receptor dimerisation. The use of statistical methods is correct and well applied to the type of data and number of variables.

In the original manuscript, they presented compelling evidence linking inhibition of gamma-secretase and p75-dependent cell death of different cell types, and also showing that oligomeric p75-CTF is less efficiently internalised and somehow protected from the gamma-secretase processing. After this revision the authors added meaningful experiments and re-oriented the model to integrate the role of TrkA signalling in endocytosis, and trying to account for the location where the processing of p75-CTF and the apoptotic signalling occur, as well as for the propagation of the apoptotic signal onboard of specialised signalling endosomes. I'm very happy with the significant expansion of the experimental data and the new shape that the article took after the revision. I'm glad to see that our comments were carefully taken onboard and led the authors to significantly improve and refine their work from the very title itself.

I have only few comments that in my opinion should be taken into account before this work is ready for publication:

1. This new orientation that includes the crucial role of TrkA-regulated internalisation of p75 is missing from the ending of the introduction where the findings are summarised.

We added a sentence reflecting this at the end of the introduction section.

2. Figure 1E has no error bars. Is not clear if the experiment was done only once. Please make sure the number of replicates is indicated for all the experiments and plots.

We added the number of replicates in all figure legends.

3. In Figure 1G-H a role of cystein 257 is argued based on the apparent decrease of cleaved caspase 3 levels that CE induces in cells expressing the C257A mutant, but this is not clear from the data, essentially for 2 reasons: i) in p75-CTF-C257A neurons cleaved caspase 3 is lower than p75-CTF-WT neurons in both with and without CE, suggesting that this decrease is independent of the role of CE; ii) the decrease is similar to what you observe when expressing p75-FL-WT and still significantly higher than EV control.

We agree with the referee that the differences between p75-CTF and p75-CTF-C257A are low, but they are significant statistically. In any case we lower the tone a bit, and change the sentence to *"The significant increment in cell death observed in p75-CTF-wt overexpressing neurons relative to the mutant, suggests a partial, but significant, contribution of the cysteine residue to the p75-CTF-mediated toxicity"*.

4. In the Experimental Procedures the conditions for maintenance of BFCN is indicated under the subtitle "Cell lines culture" which is incorrect and redundant, giving that you then describe the whole culture. In both sections the amount of NGF supplemented to the media is stated to be 100 ng, please indicate the concentration instead (I'm assuming it must say 100 ng/mL, which is rather high by the way).

We add the changes requested.

5. In the Discussion section, line 478, it is asserted that your results show that dimeric p75-CTF is not cleaved by γ -secretase "in endogenous conditions". Given that the finding were done in cell lines overexpressing a truncated receptor, it would be better to avoid that expression and replace it with something more like "in a cellular context".

We changed the sentence to *"Our studies show for the first time that γ -secretase is not able to cleave a naturally dimeric p75-CTF substrate"*.

Reviewer #2 (Comments to the Authors (Required)):

In the revised manuscript, Franco et al. have addressed all of the issues raised by this reviewer from the original version. Overall, the authors have made the revised manuscript more compelling. However, I have a few minor queries.

1. In Fig. 1I, the quality of the Western blot showing the p75-CTF is poor and appears to be cut just above the 25 kDa marker. Please provide a better quality/more convincing blot from the 3 experiments. It is also difficult to visualize the dimers in this Western blot.

The blot is not cropped and it is a full gel. We provide a higher resolution uncropped blot. Dimers of p75CTF are not visible as the gel is a normal reducing SDS-PAGE.

2. The figure legend states DR were transfected in Fig. 1I. However, the text states that cortical neurons were used for these experiments. Please clarify.

The experiment was done in cortical neurons. The mistake has been corrected in the legend

3. In Fig. 1G and 4F, how many fields of view were visualized to generate the data?
Approximately how many cells?

In the Figure 1I, cortical neurons were electroporated with a nucleofector. Every GFP+ neurons was counted in every well and condition. The experiment was done per triplicate. In total around 50 neurons were counted per condition.

In the Figure 4F, more than 500 GFP+PC12 cells were counted per condition and experiment.

This information has been added in the figure legend.

January 11, 2021

RE: Life Science Alliance Manuscript #LSA-2020-00844-TRRR

Dr. Marçal Vilar
Institute of Biomedicine of Valencia CSIC
Molecular Basis of Neurodegeneration
C/ Jaume Roig 11
València, Valencia 46010
Spain

Dear Dr. Vilar,

Thank you for submitting your Research Article entitled "TrkA mediated endocytosis of p75-CTF prevents cholinergic neurons death upon γ -secretase inhibition". It is a pleasure to let you know that your manuscript is now accepted for publication in Life Science Alliance. Congratulations on this interesting work.

DISTRIBUTION OF MATERIALS:

Again, congratulations on a very nice paper. I hope you found the review process to be constructive and are pleased with how the manuscript was handled editorially. We look forward to future exciting submissions from your lab.

Sincerely,

Shachi Bhatt, Ph.D.

Executive Editor

Life Science Alliance

<https://www.lsjournal.org/>
